# Implementation of a synthetic inflow turbulence generator in idealised WRF v3.6.1 large eddy simulations under neutral atmospheric conditions

Jian Zhong[1], Xiaoming Cai[1*] and Zheng-Tong Xie[2]

[1]School of Geography, Earth & Environmental Sciences, University of Birmingham, Edgbaston, Birmingham, B15 2TT, UK
[2]School of Engineering Sciences, University of Southampton, Southampton, SO17 1BJ, UK

*Correspondence to*: Xiaoming Cai (x.cai@bham.ac.uk)

**Abstract:** A synthetic inflow turbulence generator was implemented in the idealised Weather Research and Forecasting large eddy simulation (WRF-LES v3.6.1) model under neutral atmospheric conditions. This method is based on an exponential correlation function, and generates a series of two-dimensional slices of data which are correlated both in space and in time. These data satisfy a spectrum with a near '-5/3' inertial subrange, suggesting its excellent capability for high Reynolds number atmospheric flows. It is more computationally efficient than other synthetic turbulence generation approaches, such as three-dimensional digital filter methods. A WRF-LES simulation with periodic boundary conditions was conducted to provide *a priori* mean profiles of first- and second-moments of turbulence for the synthetic turbulence generation method and the results of the periodic case were also used to evaluate the inflow case. The inflow case generated similar turbulence structures to those of the periodic case after a short adjustment distance. The inflow case yielded a mean velocity profile and second-moment profiles that agreed well with those generated using periodic boundary conditions, after a short adjustment distance. For the range of the integral length scales of the inflow turbulence (+/-40%), its effect on the mean velocity profiles is negligible, whereas its influence on the second-moment profiles is more visible, in particular for the smallest integral length scales, e.g. with the friction velocity less than 4% error of the reference data at $x/H=7$. This implementation enables a WRF-LES simulation of a horizontally inhomogeneous case with non-repeated surface landuse patterns and can be extended so as to conduct a multi-scale seamless nesting simulation from a meso-scale domain with a km-resolution down to LES domains with metre resolutions.

**Key words:** Inflow turbulence generator, Large eddy simulation, Exponential correlation function, Atmospheric boundary layer.

## 1 Introduction

Atmospheric boundary layer flow involves a wide range of scales of eddies, from quasi two-dimensional structures at the mesoscales to three-dimensional turbulence (normally with higher Reynolds number, i.e. Re $\sim 10^8$-$10^9$) at the microscale (Muñoz-Esparza et al., 2015). The Weather Research and Forecasting (WRF) model (Skamarock and Klemp, 2008) is capable

of simulating atmospheric systems at a variety of scales. At the mesoscale and synoptic scales, the WRF model allows grid nesting for downscaling from 10-100 km to 1-10 km using a fully compressible and non-hydrostatic Reynolds-averaged Navier-Stokes (RANS) solver (Skamarock and Klemp, 2008), which captures the behaviour of mean flows only. At the microscale, a large eddy simulation (LES) can be activated in the WRF model (WRF-LES), enabling users to simulate the

characteristics of energy-containing eddies in the atmospheric boundary layer. There remain challenges in downscaling from the mesoscale (resolutions down to 1 km, capturing mean information only) to the LES scale (tens of meters or below, capturing additional turbulence information) (Doubrawa et al., 2018; Talbot et al., 2012; Chu et al., 2014; Liu et al., 2011), such as specifying the appropriate inflow conditions for an LES domain and the sub-grid scale turbulence schemes for the "gray-zone" resolution, to which neither planetary boundary layer (PBL) nor LES parametrisation schemes apply well. Consequently,

microscale and mesoscale flows are typically studied separately. Most LES models of atmospheric boundary layer flow at the microscale use periodic boundary conditions and simplified large-scale geostrophic forcing for idealised simulations. However, implicit in the use of periodic boundary conditions is the assumption that atmospheric fields and the underlying landuse have repeated periodic features. This assumption may be unrealistic for real landscapes where landuse patterns and the atmospheric phenomena coupled to them can be very heterogeneous. Therefore, such periodic WRF-LES simulations are

restricted to studies of the atmospheric boundary layer flow with a single domain (e.g. Zhu et al., 2016; Kirkil et al., 2012; Kang and Lenschow, 2014; Ma and Liu, 2017) or the outermost domain of either one-way nested cases (e.g. Nunalee et al., 2014) or two-way nested cases (e.g. Moeng et al., 2007). Here we implement a well-tested synthetic turbulence inflow scheme (Xie and Castro 2008) in the WRF-LES model (v.3.6.1), in which the meso-scale model could provide the mean flow information as the input of the synthetic turbulence inflow scheme. This scheme provides a step towards enabling WRF's

capability of nesting micro-scale turbulent flows within realistic meso-scale meteorological fields.

Dhamankar et al. (2018) reviewed three broad classes of methods to generate the turbulent inflow conditions for LES models, mainly for engineering applications. The first class is the library-based method, which relies on an external turbulence library to provide inflow turbulence. The turbulence library can be based on either: (a) the precursor/concurrent simulation (e.g.

Munters et al., 2016) on the same geometry to a main LES simulation; or (b) a pre-existing database (e.g. Schluter et al., 2004; Keating et al., 2004) from experiments or computations (on a different geometry to a main LES simulation). Although this method is usually limited to specialised applications, it can provide good-quality inflow turbulence. The second class is the recycling-rescaling based method (e.g. Lund et al., 1998; Morgan et al., 2011), in which the velocity field is recycled from some suitably selected downstream plane back to the inflow boundary plane. Although this method may be effective in

producing well-established turbulence, there are some limitations, e.g. the requirements of an equilibrium region near the inlet and a relatively large domain. The turbulence profile determined by the geometry of the precursor simulation can be added on the top of any given mean profile, which could be modified and varied in time for more realistic applications. The third class is the synthetic turbulence generator, which includes a variety of methods such as the Fourier transform-based method (e.g. Kraichnan, 1970; Lee et al., 1992), proper orthogonal-decomposition-based method (e.g. Berkooz et al., 1993; Kerschen et al.,

2005), digital-filter-based method (e.g. Xie and Castro, 2008; Klein et al., 2003; Kim et al., 2013), diffusion-based method (e.g. Kempf et al., 2005), vortex method (e.g. Benhamadouche et al., 2006) and synthetic eddy method (e.g. Jarrin et al., 2006). The synthetic turbulence generator has the potential to be used for a wide range of flows. Due to the imperfection of the synthetic turbulence, which is not directly derived from generic flow equations, these methods normally require some inputs and a certain adjustment distance for turbulence to become well-established. For more information about the above synthetic turbulence generation methods, we recommend Tabor and Baba-Ahmadi (2010), Wu (2017) and Bercin et al. (2018).

Several other methods have been developed to generate inflow turbulence for atmospheric boundary layer flow in nested WRF-LES models. Mirocha et al. (2014) introduced simple sinusoidal perturbations to the potential temperature and horizontal momentum equations near the inflow boundaries. This method can speed up the development of turbulence and generally has a satisfactory performance in the nested WRF-LES domains, providing promising results. Muñoz-Esparza et al. (2014) extended the perturbation method of Mirocha et al. (2014) and proposed four methods, i.e. the point perturbation method, cell perturbation method, spectral inertial subrange method and spectral production range perturbations, to generate perturbations of potential temperature for a buffer region near the nested inflow planes. The cell perturbation method was found to have the best performance regarding the adjustment distance for the turbulence to be fully-developed. It has the advantages of negligible computational cost, minimal parameter tuning, not requiring *a priori* turbulent information, and efficiency to accelerate the development of turbulence. Muñoz-Esparza et al. (2015) further generalised the cell perturbation method of Muñoz-Esparza et al. (2014) under a variety of large-scale forcing conditions for the neutral atmospheric boundary layer. The perturbation Eckert number (describing the interaction between the large-scale forcing and the buoyancy contribution due to the perturbation of potential temperature) was identified as the key parameter that governs the transition to turbulent flow for nested domains. They found an optimal Eckert number to establish a developed turbulent state under neutral atmospheric conditions. These methods impose temperature perturbations at specific length and time scales related to the highest resolved wave number in the LES. It was demonstrated in Muñoz-Esparza et al. (2015) that a distance of about 15 boundary-layer depths is required to allow the flow to be fully turbulent when the temperature perturbation method is adopted in the one-way nesting WRF model. Noted that the temperature perturbation method was introduced for mesoscale to microscale coupling approach where smooth mesoscale flow (no resolved turbulence) forces microscale flow by using the one-way nesting approach in WRF. Muñoz-Esparza et al. (2014) stated "the perturbation method is to provide a mechanism that accelerates the transition towards turbulence, rather than to impose a developed turbulent field at the inflow planes as the synthetic turbulence generation methods pursue", and "the use of temperature perturbations presents an alternative to the classical velocity perturbations commonly used by most of the techniques." The optimisation and generalisation of these methods would require intensive testing. Muñoz-Esparza and Kosovic (2018) extended the cell perturbation method of the inflow turbulence generation to non-neutral atmospheric boundary layers. Instead of adopting temperature perturbations in the original cell perturbation method, Mazzaro et al. (2019) further explored Random Force Perturbation Method (FCPM) in the multiscale nested domains.

Due to its accuracy, efficiency and, in particular, the capability for high Reynolds number flows, the synthetic inflow turbulence generator (Xie and Castro, 2008) has been implemented and tested on codes developed for engineering applications, such as Star-CD (Xie and Castro, 2009) and OpenFOAM (Kim and Xie, 2016), and the micro-scale meteorology code PALM

(PALM, 2017; Maronga et al., 2020). This study focuses on an implementation of this synthetic inflow turbulence generator (Xie and Castro, 2008) in the idealised WRF-LES (v3.6.1) model under neutral atmospheric conditions. In this paper, Section 2 describes the methodology of WRF-LES model and the technique of the synthetic inflow turbulence generator; Section 3 presents the results of the WRF-LES model with the use of the synthetic inflow turbulence generator; and Section 4 states the conclusions and future work.

**2 Methodology**

**2.1 WRF-LES model**

The atmospheric boundary layer is simulated by the compressible non-hydrostatic WRF-LES model, which computes large energy-containing eddies at the resolved scale directly and parameterises the effect of small unresolved eddies on the resolved field using subgrid-scale (SGS) turbulence schemes (Moeng et al., 2007). The Favre-filtered equations are (Nottrott et al.,

2014; Muñoz-Esparza et al., 2015):

$$\frac{\partial \tilde{\rho}}{\partial t} + \frac{\partial \tilde{\rho}\tilde{u}_j}{\partial x_j} = 0 \tag{1}$$

$$\frac{\partial \tilde{u}_i}{\partial t} + \frac{\partial \tilde{u}_i\tilde{u}_j}{\partial x_j} = v\frac{\partial^2 \tilde{u}_i}{\partial x_j \partial x_j} - \frac{1}{\tilde{\rho}}\frac{\partial \tilde{p}}{\partial x_i} - \frac{\partial \tau_{ij}}{\partial x_j} + \tilde{F}_i, \tag{2}$$

where $i$ (or $j$) = 1, 2, 3, represents the component of the spatial coordinate, $\tilde{u}_i$ is the filtered velocity, $x_i$ is the spatial coordinate, $t$ is the time, $\tilde{p}$ denotes the filtered pressure, $\tilde{\rho}$ is the filtered density, $v$ is the fluid kinematic viscosity, $\tau_{ij}$ are the SGS stresses,

$\tilde{F}_i$ represents external force terms (normally involving the Coriolis force caused by the rotation of the Earth and the large-scale geostrophic forcing).

For the closure of Eq. (2), $\tau_{ij}$ are parameterised using a SGS model. In this study, the 1.5-order turbulence kinetic energy (TKE) SGS model is used,

$$\tau_{ij} = -2v_{sgs}\tilde{S}_{ij}, \tag{3}$$

where $\tilde{S}_{ij}$ is the filtered strain-rate tensor and calculated as,

$$\tilde{S}_{ij} = \frac{1}{2}\left(\frac{\partial \tilde{u}_i}{\partial x_j} + \frac{\partial \tilde{u}_j}{\partial x_i}\right), \tag{4}$$

$v_{sgs}$ denotes the SGS eddy-viscosity and is defined as,

$$v_{sgs} = C_k \ell k_{sgs}^{1/2}, \tag{5}$$

where $C_k$ is a model constant, $\ell$ is the SGS length scale, which equals the grid volume of size ($\Delta$) under neutral conditions (Deardorff, 1970),

$$\Delta = (\Delta x\, \Delta y \Delta z)^{1/3}, \tag{6}$$

$k_{sgs}$ is the SGS TKE with the transport equation

$$\frac{\partial k_{sgs}}{\partial t} + \frac{\partial}{\partial x_i}\left(k_{sgs}\tilde{u}_i\right) = -\frac{v_{sgs}}{Pr}\frac{g}{\theta_0}\frac{\partial \tilde{\theta}}{\partial z} + 2v_{sgs}\tilde{S}_{ij}\tilde{S}_{ij} + \left(v + v_{sgs}\right)\frac{\partial^2 k_{sgs}}{\partial x_i \partial x_i} - C_\varepsilon \frac{k_{sgs}^{1.5}}{\ell}, \tag{7}$$

where $\tilde{\theta}$ is the filtered potential temperature, $Pr$ is the turbulent Prandtl number, and $C_\varepsilon$ is a dissipation coefficient (for more details about the parameterisation see Moeng et al. (2007)). Without loss of generality, the " $\tilde{\phantom{x}}$ " notation for all filtered

variables is omitted hereafter.

## 2.2 Synthetic inflow turbulence generator

The synthetic inflow turbulence generator in Xie and Castro (2008) adopted the digital filter-based method and is used in this study. For simplicity, a one-dimensional problem (the streamwise velocity, $u$, along the $x$-direction) is used as an illustration to describe this method. The two-point velocity correlations $R_{uu}(k\Delta x)$ are assumed to be represented by an exponential

function:

$$\frac{\overline{u_m u_{m+k}}}{\overline{u_m u_m}} = R_{uu}(k\Delta x) = \exp\left(-\frac{\pi|k|}{2n}\right), \tag{8}$$

where $m$, the index that the averaging operator is applied, denotes the $m$-th element of a vector (one-dimensional data series of, for example, the digital-filtered velocity, $u$, in (9) below), $k$ is the number of elements for the two-point distance of $k\Delta x$, $n$ is related to the integral length scale $L = n\Delta x$ with the grid size of $\Delta x$, $u_m$ is the digital-filtered velocity,

$$u_m = \sum_{k=-N}^{N} b_j r_{m+k}, \tag{9}$$

where $r_m$ is a sequence of random data with mean $\overline{r_m} = 0$ and variance $\overline{r_m r_m} = 1$, $N$ is related to the length scale for the filter (here $N \geq 2n$), and $b_j$ is the filter coefficient and can be estimated from

$$b_k = \tilde{b}_k / \left(\sum_{j=-N}^{N} \tilde{b}_j^2\right)^{1/2}, \text{ where } \tilde{b}_k \cong \exp\left(-\frac{\pi|k|}{n}\right). \tag{10}$$

For a two-dimensional filter coefficient, it can be obtained that

$$b_{jk} = b_j b_k, \tag{11}$$

which will then be used to filter the two-dimensional random data at each time step,

$$\varphi_\beta(t, x_j, x_k) = \sum_{j=-N_j}^{N_j} \sum_{k=-N_k}^{N_k} b_{jk} r_{m+j, m+k}, \tag{12}$$

where $\beta$ indicates the velocity component. At the next time step, the filtered velocity field is calculated as,

$$\Psi_\beta(t + \Delta t, x_j, x_k) = \Psi_\beta(t, x_j, x_k) \exp\left(-\frac{\pi \Delta t}{2T}\right) + \varphi_\beta(t, x_j, x_k)\left[1 - \exp\left(-\frac{\pi \Delta t}{T}\right)\right]^{0.5}, \tag{13}$$

where $T$ is the Lagrangian time scale representing the persistence of the turbulence, $\varphi_m(t, x_j, x_k)$ is calculated based on Eq.

(12). Xie and Castro (2008) demonstrated that Eq. (13) satisfies the correlation functions in an exponential form in space and in time. The two-dimensional filter in Xie and Castro (2008) is more computationally efficient than a three-dimensional filter. Finally, the velocity field is obtained by using the simplified transformation proposed by Lund et al. (1998),

$$\tilde{u}_i = \bar{u}_i + \alpha_{i\beta} \Psi_\beta, \tag{14}$$

where

$$[\alpha_{i\beta}] = \begin{bmatrix} \left(\tilde{R}_{11}\right)^{1/2} & 0 & 0 \\ \tilde{R}_{21}/\alpha_{11} & \left(\tilde{R}_{22} - (\alpha_{21})^2\right)^{1/2} & 0 \\ \tilde{R}_{31}/\alpha_{11} & (\tilde{R}_{32} - \alpha_{21}\alpha_{31})/\alpha_{22} & \left(\tilde{R}_{33} - (\alpha_{31})^2 - (\alpha_{32})^2\right)^{1/2} \end{bmatrix}, \tag{15}$$

and $\tilde{R}_{i\beta}$ is the resolved Reynolds stress tensor, which can be estimated based on measurements or other simulations with periodic boundary conditions. The calculations of $\alpha_{i\beta}$ follow an iterative order: $\alpha_{11}$, $\alpha_{21}$, $\alpha_{22}$, $\alpha_{31}$, $\alpha_{32}$, and $\alpha_{33}$.

### 2.3 Model coupling and configuration

In this study, we firstly configured a WRF-LES model with periodic boundary conditions in both the streamwise and spanwise
directions to obtain *a priori* mean profiles of first- and second-moments of turbulence, such as the vertical profiles of mean velocity and Reynolds stress components, which are required as input by the synthetic inflow turbulence generator. Additional essential quantities as input of the inflow generator are three integral length scales in the *x, y* and *z* directions, denoted by $L_x$, $L_y$ and $L_z$, respectively (or $L_i$, *i=x,y,z*). For the inflow BASE case (denoted by LS1.0), the vertical profiles of $L_i$ are specified as functions of $z/H$, where $H$ is the boundary layer height (500 m in this study), shown as Fig. 1, similar to those in Xie and
Castro (2008). The streamwise length scale ($L_x$) is specified based on the mean streamwise velocity profile ($\langle u \rangle$) and a constant Lagrangian time scale $T$ (prescribed in Eq. 13), i.e. $L_x = T\langle u \rangle$ using Taylor's hypothesis (turbulence is assumed to be frozen while it is moving downstream with a mean speed of $\langle u \rangle$). The spanwise length scale ($L_y$) is specified as a constant value. The vertical length scale ($L_z$) is specified as a smaller constant value near the bottom and a larger constant value for the upper domain to be closer to the measured length scales, as explained in Xie and Castro (2008) and Veloudis et al. (2007). We
conducted a sensitivity study of integral length scales by varying all three baseline $L_x$, $L_y$ and $L_z$ with the ratio of 0.6, 0.8, 1.0, 1.2, or 1.4; these individual cases are denoted by "LS0.6", "LS0.8", "LS1.0", "LS1.2", "LS1.4", respectively, in which "LS1.0" is the base case. The size of the computational domain is 9.98 km×2.54 km×0.5 km (in the *x*, *y* and *z* directions), with the resolutions of $\Delta x = \Delta y = 20\ m$ and stretched $\Delta z$ (from about 3 m up to 27 m). The grid number is then 499×127×49. In order

to achieve the constant wind direction vertically, the Coriolis force is not activated in this study. The external driving force is specified as a constant pressure gradient force in Eq. (2) , similar to that used in Ma and Liu (2017), resulting in a prevailing wind speed of about 10 m s$^{-1}$ at the domain top. At the top boundary, a rigid lid ("top_lid" in the "namelist.input" file of the WRF-LES model) is specified, and a Rayleigh damping layer of 50 m is used to prevent undesirable reflections (Nottrott et

al., 2014; Ma and Liu, 2017) and to maintain a neutral atmospheric boundary layer.

For the cases with the synthetic turbulence at the inlet and periodic conditions in the spanwise direction, the constant pressure gradient force is not necessary anymore. Instead, a pressure-drop between the inlet and outlet is implicitly derived from the prescribed mean momentum profiles as part of the synthetic inflow and the outflow boundary conditions in the solver. The

periodic case is used for the validation of the results from the inflow case. The WRF-LES is solved at a time step of 0.2 s. A spin-up period of 6 h is adopted for all inflow cases to allow turbulence inside the domain to reach quasi-equilibrium. The further 1 h outputs with 5 second interval (approximately the advection timescale of the smallest resolved eddies, which is equivalently twice the grid resolution of 20 m) were used for the analysis. We take advantage of the homogeneous turbulence in the spanwise direction (Ghannam et al., 2015) by calculating all resolved-scale turbulent quantities by averaging in the

spanwise (the $y$-direction) direction and in time $t$ over the last 1 h period. This averaging is referred to as "the $y$-$t$ averaging" hereafter, and is denoted by $\langle \varphi \rangle$, for example, for the $y$-$t$ averaged $\varphi$. For a 4D variable, $\varphi(t, x, y, z)$, the $y$-$t$ averaged $\varphi$ is a function of $x, z$, i.e. $\langle \varphi \rangle(x, z)$; for a variable defined on the $x$-$y$ plane, e.g. friction velocity $u_*(t, x, y)$, the $y$-$t$ averaging $u_*$ is a function of $x$, i.e. $\langle u_* \rangle(x)$.

In the synthetic inflow turbulence generator, a uniform mesh is used with resolutions of $\Delta y = 20\ m$ (same as that on the

physical inlet of the WRF-LES domain) and $\Delta z = 4.2\ m$ (slightly larger than the smallest vertical grid spacing of the WRF-LES domain). The three filtered velocity components at the inlet from the inflow generator are then interpolated onto the vertically non-uniform mesh in the WRF-LES domain. It should be noted that the grid resolution can differ between the inflow patch and the inlet of the WRF-LES domain. The standalone synthetic turbulence generator code in Xie and Castro (2008) was originally run on a single processor, whereas the WRF-LES simulation here is run in parallel mode. It is therefore

necessary to ensure that each processor in the parallel mode has the same information of the 2-dimensional slice of flow field before each processor can extract the corresponding patch from the same 2-dimensional inlet data. In this implementation, the synthetic turbulence generator code is firstly run on the master processor at each WRF-LES time step. The generated inlet data are then passed to other processors. The flow field at the inlet of each corresponding processor was then be updated at every time step. The additional computational time for the inflow case is associated with the synthetic inflow turbulence generator

and data passing, i.e. non-parallelisation of the current inflow generator. Increasing the integral length scale would increase the computation time since bigger arrays are constructed and computed for the filtered velocity in the synthetic inflow turbulence generator, as in Eq. (9) for the larger integral length scale.

## 3 Results

### 3.1 BASE case output

#### 3.1.1 Horizontal slices of instantaneous streamwise velocity component

Figure 2 illustrates the horizontal slices of the instantaneous streamwise velocity component at $z/H = 0.1$ in the periodic case, the synthetic inflow case (LS1.0 in Fig. 1a), and the inflow case without inlet perturbations (with mean information only) after 6 hours' simulation time. The synthetic turbulence structures imposed at the inlet are advected into the domain, and are adjusted by the model dynamics at further downwind distances. After an adjustment distance (about $x/H = 5$-10), the inflow case (LS1.0) clearly generates turbulence streaks, which are similar to these in the periodic case. Other quantities that may further demonstrate this adjustment distance will be discussed in the following subsections. This suggests that the synthetic turbulence generated at the inlet can develop into realistic turbulence with well-configured structures from an adjustment distance downwind of about $x/H = 5$-10. For the inflow case without inlet velocity perturbations, there is almost no turbulence generated in the domain even after several hours of simulation. This is consistent with other similar tests using engineering CFD codes with no synthetic turbulence added at the inlet, e.g. Xie and Castro (2008), which confirms that a very long distance (e.g. 100 times boundary layer thickness) is needed to allow turbulence to develop. This indicates the importance of imposing synthetic turbulence, or at least some form of random perturbations (e.g. Muñoz-Esparza et al., 2015) at the inlet. The inflow case without the inlet velocity perturbations is not presented in the later sections.

#### 3.1.2 Development of local friction velocity

Figure 3 shows the development of the *y-t* averaged local friction velocity, $\langle u_* \rangle(x)$, for the periodic case and the inflow BASE case (LS1.0), normalised by $u_*$, the *x-y-t*-averaged friction velocity for the periodic case. The variation of the local friction velocity is within ±0.5% $u_*$ along the streamwise direction for the periodic case and is within 1.5% $u_*$ for the inflow case after a downwind distance of $x/H =$ 7. There is a larger variation close to the inlet region ($x/H < 7$) for the inflow case. This is because the imposed turbulence on the inflow plane is 'synthetic', of which only the first order and second order moments, integral length scales and the spectra aim to match the prescribed data (Bercin et al., 2018). It must develop over a certain distance in the WRF-LES domain before it can be fully developed 'realistic' turbulence.

#### 3.1.3 Horizontal profiles of mean flow and turbulence quantities

Figure 4 illustrates the *y-t* averaged horizontal profiles of the normalised mean streamwise velocity component, normal and shear turbulent stresses, and TKE at $z/H = 0.1$ and $z/H = 0.5$ for the periodic case and the inflow case (LS1.0), respectively. These horizontal profiles show the development of synthetic turbulence along the streamwise direction. There are only slight differences in the normalised mean streamwise velocity component ($\langle u \rangle /u_*$) between the periodic case and the inflow case. This suggests that the inflow case reproduces successfully the desired mean wind. The curves of normalised streamwise

velocity variance ($\langle u'^2 \rangle / u_*^2$) for both cases match well with each other downstream from $x/H = 7$-8, although there is a sudden jump close to the inlet and a subsequent decrease until the location of convergence. The horizontal profiles of normalised cross-stream velocity variance ($\langle v'^2 \rangle / u_*^2$) for the inflow case are in a good agreement after a developing distance of $x/H = 10$-12, compared with those for the periodic case. The development convergence of normalised vertical velocity variance ($\langle w'^2 \rangle / u_*^2$) is achieved after a distance of about $x/H = 5$-10 from the inlet. The development distance of turbulent shear stress ($\langle u'w' \rangle / u_*^2$) is about $x/H = 5 - 15$. Since the streamwise velocity variance comprises a large proportion of TKE, the development distance for TKE is similar to that for the streamwise velocity variance, i.e. about $x/H = 7$-8. The distance needed for different quantities to reach a converged state differs from each other, and it is about $x/H = 5$-15.

### 3.1.4 Vertical profiles of mean flow and turbulence quantities

Figure 5 shows the *y-t* averaged vertical profiles of the normalised mean streamwise velocity component, normal and shear turbulent stresses, and TKE at a series of downwind locations, $x/H = 0, 4, 6$ and 10, for the inflow case (LS1.0). Inflow cases are not averaged in the streamwise direction so that the development of turbulence at each downwind location ($x/H$) can be investigated. Red lines in Fig. 5 are the spatially (including both in the streamwise and spanwise directions) and temporally averaged vertical profiles for the periodic case. It is noted again that these data for the periodic case are also used as the inputs for *a priori* turbulence information required by the synthetic inflow turbulence generator. It is also noted that the profiles of the mean velocity and second order moments at the inlet ($x/H = 0$) are overall in a good agreement with these of the periodic case, which suggests precise settings of the turbulence generator. The profiles of the normalised mean streamwise velocity component ($\langle u \rangle / u_*$) in the inflow case match closely those of the periodic case. Although the sampled data are limited, this confirms again that the inflow case achieves the desired the mean wind. The normalised streamwise velocity variance ($\langle u'^2 \rangle / u_*^2$) converges towards the periodic profile after $x/H = 6$ as shown in Fig. 5 (b). Although the vertical profiles of $\langle v'^2 \rangle / u_*^2$, $\langle w'^2 \rangle / u_*^2$ and $\langle TKE \rangle / u_*^2$ for the inflow case show small variations between different locations, they are all in a good agreement with the corresponding data of the periodic case. These are consistent with the results shown in Fig. 4. The turbulent shear stress $\langle u'w' \rangle$, which is the cross correlation between the streamwise and vertical velocity fluctuations, usually converges more slowly than the normal turbulent stresses, e.g. $\langle v'^2 \rangle$. Overall, the synthetic inflow turbulence generator performs well in terms of the mean flow and the turbulence quantities against the data from the periodic case, as well as the short development distance.

### 3.1.5 Spectral analysis

Figure 6 illustrates the spectra of the streamwise velocity component at a series of downwind locations ($x/H = 0, 4, 6$, and 10) at $z/H = 0.5$ for the periodic case and the inflow case (LS1.0). For each *x*-location, e.g. $x/H = 10$, the spectrum for the inflow case was first calculated from the streamwise velocity component over a time series of 3600 s with an interval of 5 s

for five selected sample locations of $y_n$ ($y/H = 1.76, 2.16, 2.56, 2.96$ and $3.36$), namely, $\tilde{u}(t, 2H, y_n, 0.5H)$. The spectral data were then averaged over $y_n$ to give the spectra plotted in Fig. 6.

The spectrum for the periodic case is calculated using the same method as that used for the inflow case, with an additional average over the streamwise direction $x$. The spectrum at the inlet ($x/H$=0) possesses the broadest range of wavenumbers where eddies exhibit inertial sub-range behaviour, as evidenced by the wavenumber range within which the slope of each spectrum is approximately -5/3. There is an evidence of the tendency in the profiles from the inlet downstream to recover to that of the periodic case. The spectrum drops slightly at high wavenumbers from the imposed spectra at $x/H = 0$ to downwind locations, and approaches the spectrum of the periodic case. The slight drop suggests a decay of small eddies due to the SGS viscosities and the numerical dissipation originating from the advection scheme in the WRF-LES model. The spectra in Muñoz-Esparza et al. (2015) drop at lower wave numbers than those in Fig. 6, mainly due to a coarser resolution (than the current one). Our resolution of 20 m in the horizontal direction is much finer than the resolution of 90 m in Muñoz-Esparza et al. (2015). In other words, the size of the smallest eddy (twice the grid resolution) that can be resolved by the LES model is 40 m in our paper vs 180 m in Muñoz-Esparza et al. (2015).. These confirm that synthetic turbulence with an inertial subrange in the spectrum generated by using Xie and Castro (2008) method is able to be mostly sustained in WRF-LES for a high resolution. It is noted that for a very high resolution, e.g. in the order of magnitude 1 meter, similar as that used in the simulations of PALM (PALM, 2017), the inertial subrange in the spectrum is much wider. It is to be noted that Muñoz-Esparza et al. (2015) also tested the Xie and Castro (2008) method in WRF-LES using the same resolution of 90 m as that for the temperature perturbation method. Again this is rather a coarse resolution to test the performance of the Xie and Castro (2008) method when a spectrum is of interest.

**3.2 Sensitivity tests of integral length scale in the flow cases**

It is not trivial to obtain 'accurate' integral length scales of the inlet turbulence generator. Indeed these data are always incomplete. Therefore, it is necessary to conduct sensitivity tests of the integral length scales. Figure 7 shows the influence of integral length scale on the development of local friction velocity. Various integrate length scale ratios (ranging from 0.6 to 1.4) to those ($L_x$, $L_y$ and $L_z$ respectively) in the LS1.0 case are tested. Note that these three integral length scales ($L_x$, $L_y$ and $L_z$ ) are in the same ratio to those respectively in the LS1.0 case. For all inflow cases, there is a sudden change near the inlet due to the imposed 'imperfect' inflow turbulence. The adjustment distance to well-established turbulence (i.e. within 4 % error) is generally short, i.e. about $x/H = 2$-$7$ for the studied cases LS0.6-1.4, but seems shorter for the case with the smaller integral length scales. This suggests that the imposed integral length scales for the inflow turbulence slightly affect the convergence to well-developed turbulence. We conclude that a variation of ±40% in the integral length scale in the cases LS0.6-1.4 yields a variation of less than 4% in the local friction velocity after about $x/H = 7$, and that the sensitivity of integral

length scale on the local friction velocity is not significant in the WRF-LES model if the used integral length scale is within a reasonable range. This is consistent with that in engineering type CFD solvers in Xie and Castro (2008).

Figure 8 shows the effects of integral length scale on the horizontal profiles of the normalised mean streamwise velocity, normal and shear turbulent stresses, and TKE at $z/H = 0.5$. Figure 8 (a) shows that $\langle u \rangle / u_*$ is slightly greater for the length scale ratio less than 1.0. This is likely due to a slightly smaller $u_*$, which is common for smaller integral length scale cases (as shown in Fig. 7). Figures 8 (b-d) and (f) show that in general the normal stresses, $\langle u'^2 \rangle / u_*^2$, $\langle v'^2 \rangle / u_*^2$, $\langle w'^2 \rangle / u_*^2$, and $\langle TKE \rangle / u_*^2$ increase as the length scale ratio increases. This is because small eddies tend to decay faster than large eddies. It is crucial to note that for those with the integral length scales close to those of LS1.0 (the base case) the development distance to converged turbulence is shorter compared to other cases, indicating that the length scales of the base case are reasonable estimations.

Figure 9 shows effects of integral length scale on vertical profiles of the mean velocity, normal and shear turbulent stresses and TKE at a typical streamwise location ($x/H = 10$). These profiles are consistent with those in Fig. 8 and draw the same conclusions as from Fig. 8. For all the tested integral length scales, downstream from $x/H = 10$ both mean and turbulent quantities converge to the periodic case. This suggests again that the mean velocity and the turbulent stresses are not very sensitive to the integral length scales if they are not too different from the realistic values. In general, there are slight differences in $\langle u \rangle / u_*$ between each case. The magnitudes of turbulent quantities for smaller integral length scales are generally smaller than those for larger integral length scales.

Figure 10 shows the effect of the integral length scale on the spectra of the streamwise velocity component at $x/H = 10$ and $z/H = 0.5$. For all cases tested in the current study, the spectra with various integral length scales generally match those of the periodic case at a distance of $x/H = 10$ from the inlet albeit with slight changes of the spectrum for small wavenumber turbulence. A very small variation of the spectra is within margins of uncertainty in the calculation of the spectra from the raw data. All spectra show an inertial subrange of -5/3 slope, which are consistent with those in the references, such as Xie and Castro (2008), indicating of the robustness of the synthetic turbulence generator on the generation of an inertial subrange.

**4 Discussion and conclusions**

A synthetic inflow turbulence generator (Xie and Castro, 2008) was implemented in an idealised WRF-LES (v3.6.1) model under neutral atmospheric conditions. A WRF-LES model with periodic boundary conditions was firstly configured to provide *a priori* turbulence statistical data for the synthetic inflow turbulence generator. Previous studies (e.g. Xie and Castro, 2008) suggest that it is important to have an approximation of the integral length scales, which are the key inputs of the inflow turbulence generator. The results from the inflow cases were then compared with those from the periodic case. It is important

to estimate the integral length scales, which are the key inputs of the inflow turbulence generator. Therefore sensitivity tests were conducted for the response of the local friction velocity, the mean flow, the Reynolds stresses, and the turbulence spectra for the flow cases for varying integral length scales.

The inflow case with the baseline integral length scales generates similar turbulence structures to those for the periodic case after an adjustment distance of $x/H = 5$-15. The WRF-LES model with the inflow generator reproduces realistic features of turbulence in the neutral atmospheric boundary layer. The development of local friction velocity suggests that a downwind distance of about $x/H = 7$ is required to recover the local friction force for the inflow case, which agrees with the findings in Xie and Castro (2008) and Kim et al (2013). Keating et al. (2004) suggested a development distance of about 20 half-channel

depth for modelling a plane channel flow. The difference between this value and our results can be attributed to the different synthetic turbulence generation approaches adopted here versus those adopted by Keating et al. (2004). Laraufie et al. (2011) suggested that an increase in the Reynolds number decreases the adjustment distance when a synthetic inflow turbulence generator is used. For our simulated atmospheric boundary layer flow here, the Reynolds number is extremely large. Thus adopting synthetic inflow turbulence generator for the atmospheric boundary layer should also be advantageous in engineering

applications. Regarding the minimum resolution required to generate turbulence synthetically, our presented results confirm that the tested grid resolution sufficiently resolves the important features.

Horizontal and vertical profiles of mean velocity and second-moment statistics further confirm that a short adjustment distance is required for the development of synthetic turbulence. The mean velocity profiles at all tested locations were in very good

agreement with the reference data, while the turbulence second moment statistics profiles were in reasonable agreement with the reference data about $x/H = 5$-15 downwind of the inlet. An accurate estimation of the second order moments are crucial for the assessment of the synthetic inflow turbulence generator, in particular when the inflow turbulence information is not completely available. We found varying the integral length scale within +/-40% of the value in the base case has a negligible influence on the mean velocity profiles, while the effects of the variation on the turbulent second order moment statistics are

visible, for example the local friction velocity was within 4 % error of the reference data at $x/H=7$. The synthetic inflow turbulence generator requires additional computational time compared to periodic boundary conditions. This will be certainly improved by running the synthetic inflow generation subroutine in parallel as a future task. This study is focused on the feasibility of implementing the inflow method (Xie & Castro, 2008) in the meso-to-micro-scale meteorological code WRF and the impact of the key variables (i.e. the integral length scales) on the simulated turbulence development inside the domain.

This inflow subroutine has previously been implemented in both serial and parallel mode in several codes, including engineering type of codes Star-CD (Xie and Castro, 2009) and OpenFOAM (Kim and Xie, 2016), and the micro-scale meteorology code PALM (PALM, 2017). Although the current implementation in WRF is affordable for a moderate-sized simulation (e.g. tens of meters resolutions), the technical parallelisation of this inflow subroutine in WRF-LES can be the

future work for very large simulation domains with high resolutions. Users of our open source subroutine may offer this technical contribution.

In summary, the synthetic inflow turbulence generator is implemented successfully into the idealised WRF-LES model. The

generated two-dimensional slices of data are correlated both in space and in time in the exponential form. The spectrum of these data shows an inertial subrange of -5/3 slope, and this again suggests the capability of the method to generate high Reynolds number flows. These tests on WRF also confirm that this method yields a satisfactory accuracy, after having compared the local friction velocity, the mean velocity, the Reynolds stresses and the turbulence spectra against the reference data. The WRF-LES model with the synthetic turbulence generator provides promising results as evaluated against the periodic

case. The limitation of this method is the requirement of *a priori* turbulence statistic data and integral length scales, which can be estimated by the similarity theory of the atmospheric boundary layer or experimental data. Sensitivity studies have been performed to address this issue, in particular in terms of effect of the integral length scale. The implementation of the synthetic inflow turbulence generator (Xie and Castro, 2008) can be extended to the WRF-LES simulation of a horizontally inhomogeneous case with non-repeated surface land-use patterns, and be further developed for the multi-scale seamless nesting

case from a meso-scale domain with a km-resolution down to LES domains with metre resolutions. It is also worth a future study to examine the wind spiral case induced by the Coriolis force in the atmospheric boundary layer.

*Code and data availability.*

The standard version of WRF v3.6.1 is available at http://www2.mmm.ucar.edu/wrf/users/download/get_sources.html. The

coupling WRF v3.6.1 code with the synthetic inflow turbulence generator and case settings are archived on Zenodo (https://doi.org/10.5281/zenodo.3668352).

*Author contributions.*

The study was conceived by XC and ZX; JZ implemented the synthetic inflow turbulence generator code (from ZX) to the

WRF-LES model (v3.6.1) and ran model simulations; all authors contributed to writing the manuscript.

*Competing interests.*

The authors declare that they have no conflict of interest.

*Acknowledgements.*

This work is part of the CityFlocks project, sponsored by the UK Natural Environment Research Council (NERC: NE/N003195/1). This work used the ARCHER UK National Supercomputing Service (http://www.archer.ac.uk).

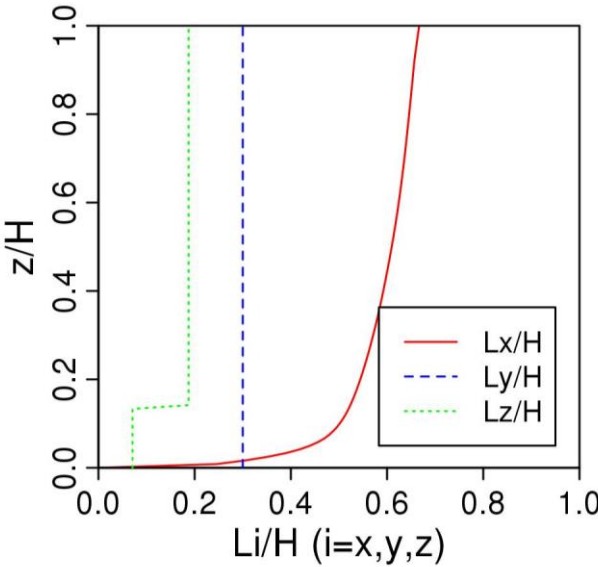

Figure 1: Integral length scales prescribed at the inlet used in the inflow BASE case (LS1.0).

30

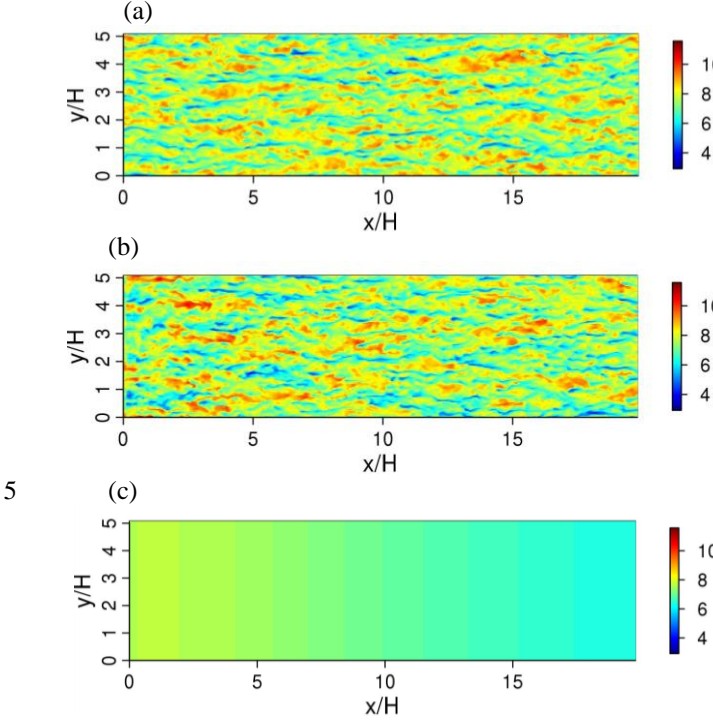

Figure 2: Horizontal slice of instantaneous streamwise velocity component, $u$ (m s$^{-1}$), at $z/H$=0.1 after 6 hours' simulation: (a) the fully periodic case, (b) the synthetic inflow BASE case (LS1.0), and (c) the inflow case without perturbations at the inlet.

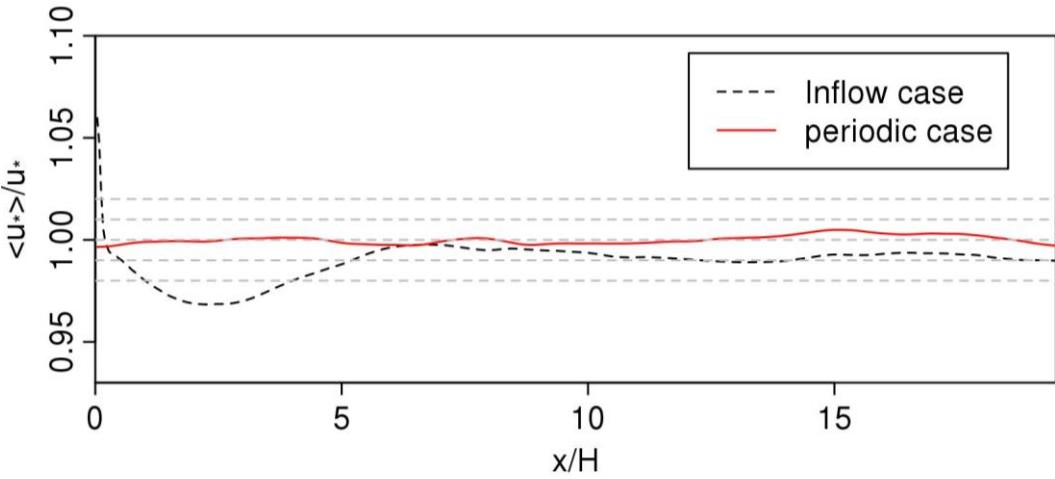

Figure 3: Spatial variation of $\langle u_* \rangle / u_*$ for the periodic case and the inflow case (LS1.0), where $\langle u_* \rangle$ is the $y$-$t$ averaged local friction velocity and $u_*$ is the $x$-$y$-$t$-averaged friction velocity for the periodic case.

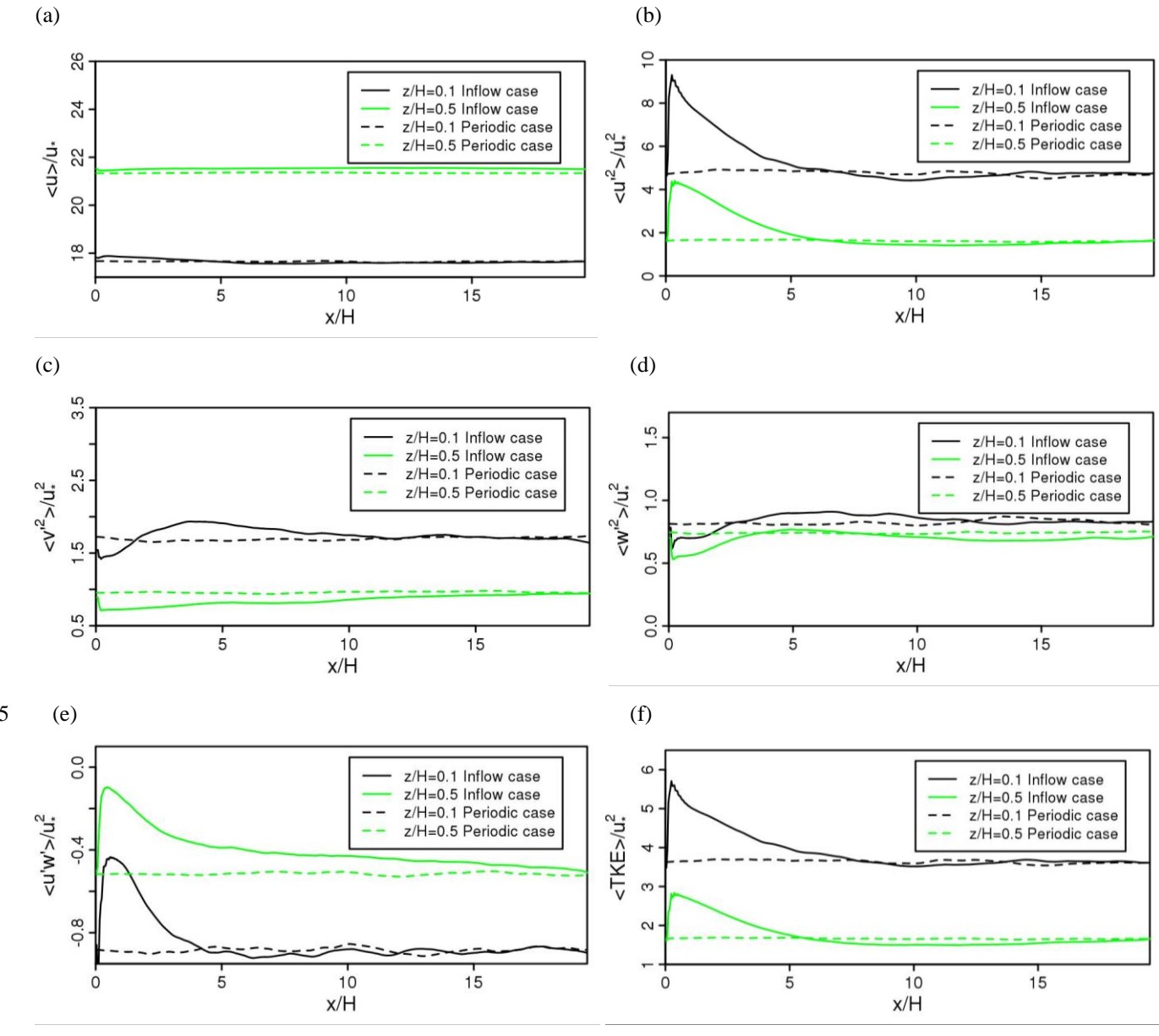

Figure 4: Horizontal profiles (spatially and temporally averaged) of (a) $\langle u \rangle / u_*$, (b) $\langle u'^2 \rangle / u_*^2$, (c) $\langle v'^2 \rangle / u_*^2$, (d) $\langle w'^2 \rangle / u_*^2$, (e) $\langle u'w' \rangle / u_*^2$, and (f) $\langle TKE \rangle / u_*^2$ at z/H=0.1 and z/H=0.5 in the periodic case and the inflow case (LS1.0).

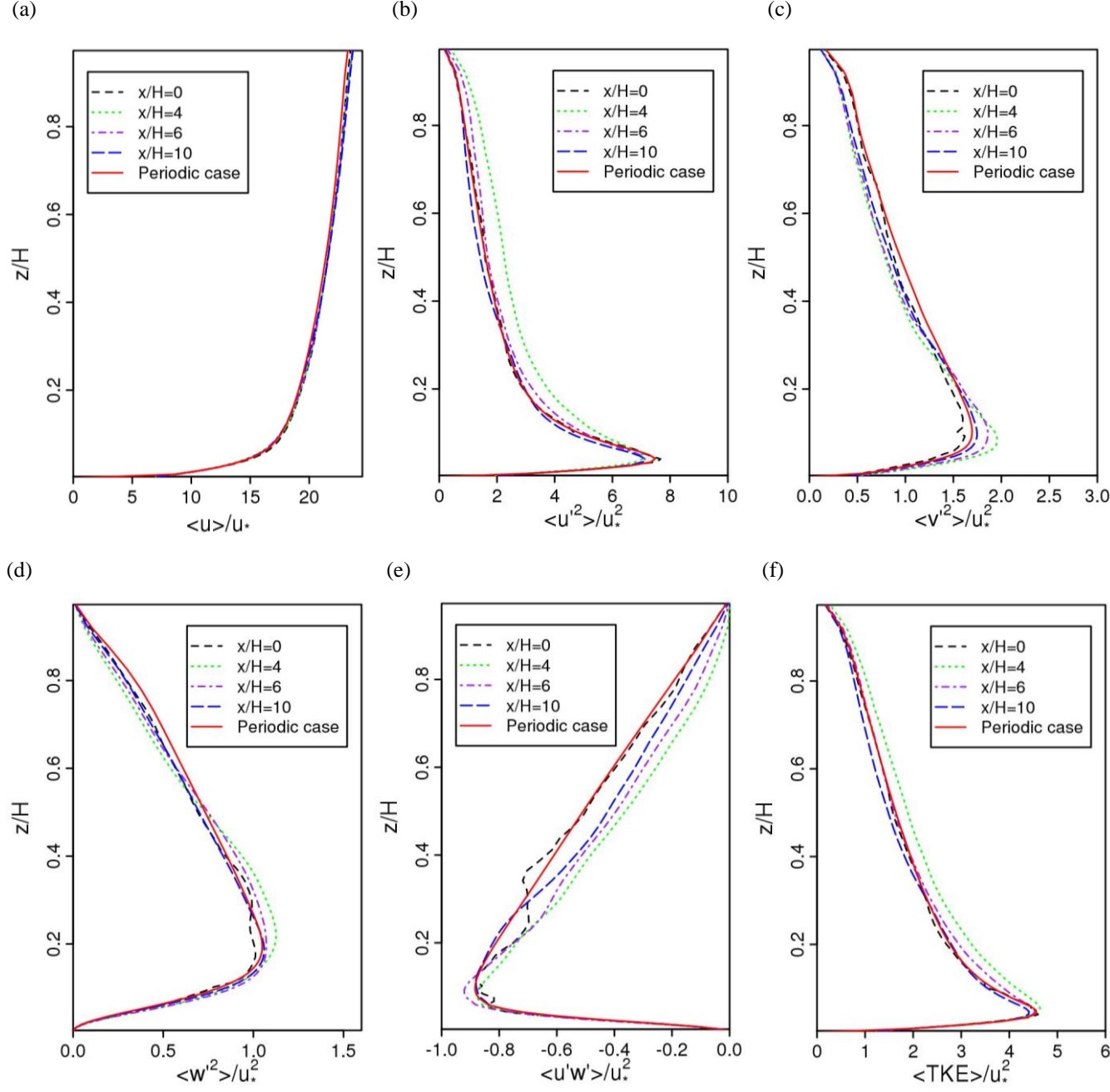

5  Figure 5: Spatially and temporally averaged vertical profiles of (a) $\langle u \rangle / u_{**}$, (b) $\langle u'^2 \rangle / u_*^2$, (c) $\langle v'^2 \rangle / u_*^2$, (d) $\langle w'^2 \rangle / u_*^2$, (e) $\langle u'w' \rangle / u_*^2$, and (f) $\langle TKE \rangle / u_*^2$ at a series of downwind locations in the inflow case (LS1.0), and the periodic case (also averaged in the streamwise direction).

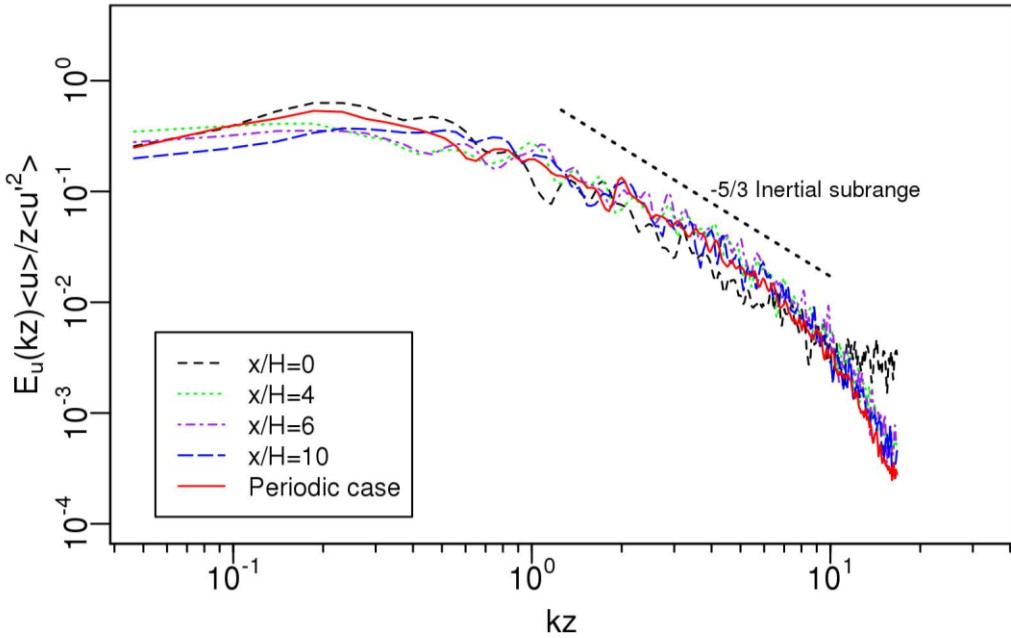

Figure 6: Spectra of streamwise velocity component for a series of downwind locations at the height of $z/H$=0.5, $k$ is the angular wavenumber, with $\langle u \rangle$ and $\langle u'^2 \rangle$ the spatially averaged mean and streamwise normal turbulent stress, respectively.

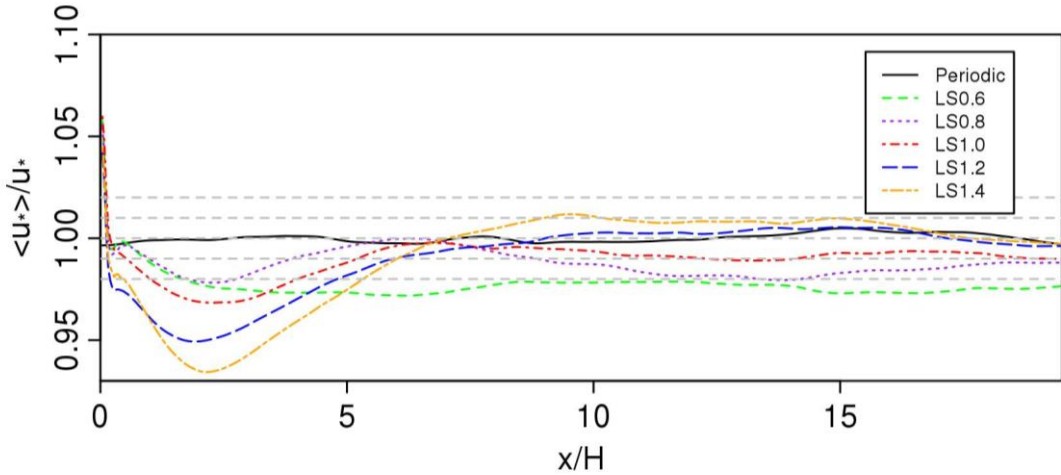

Figure 7: Development of local friction velocity (averaged over spanwise direction and time) with various integral length scales. $\langle u_* \rangle$ is the local friction velocity along the streamwise direction, and $u_*$ is the x-y-t-averaged friction velocity for the periodic case.

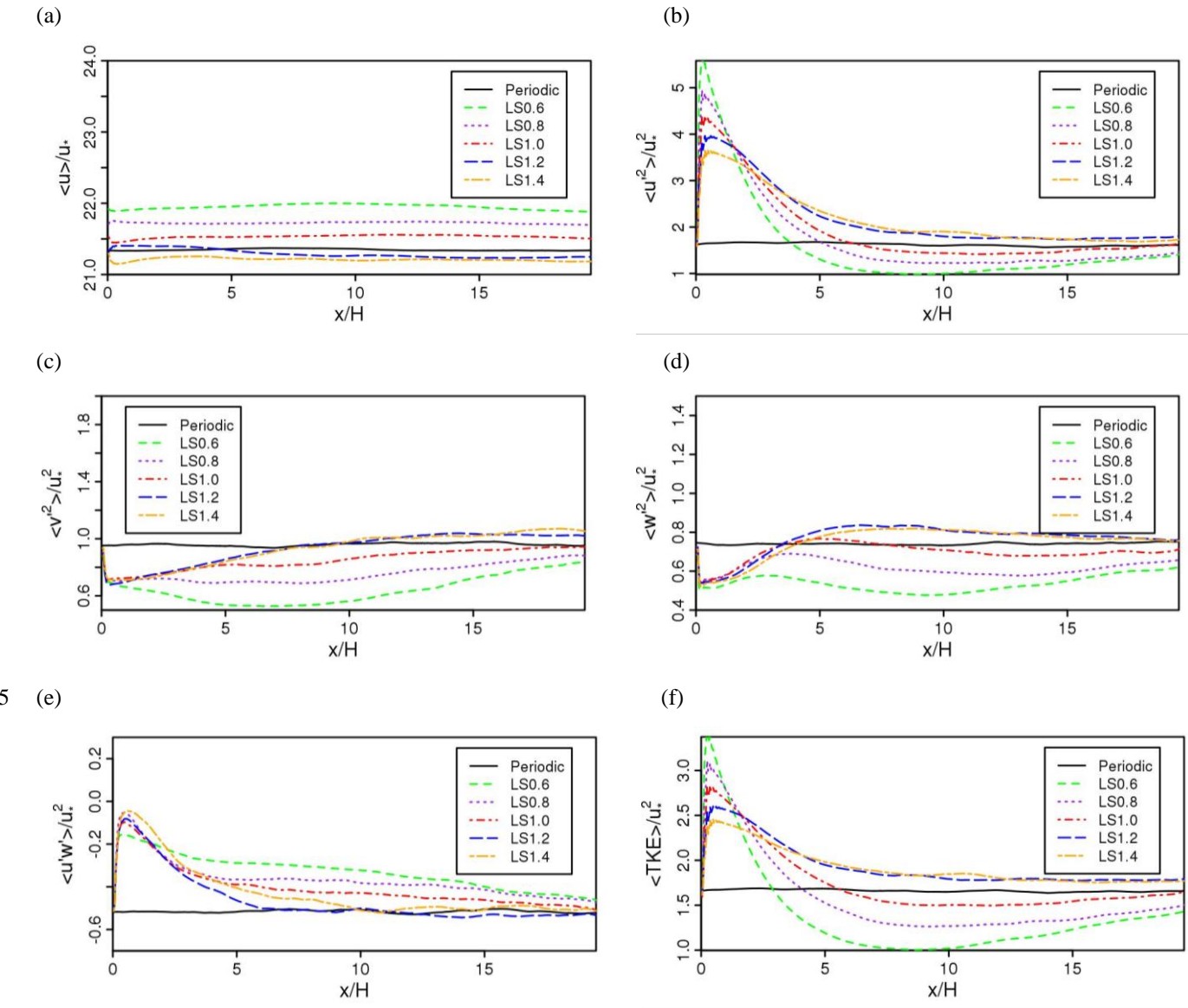

Figure 8: Horizontal profiles (spatially and temporally averaged) of (a) $\langle u \rangle / u_*$, (b) $\langle u'^2 \rangle / u_*^2$, (c) $\langle v'^2 \rangle / u_*^2$, (d) $\langle w'^2 \rangle / u_*^2$, (e) $\langle u'w' \rangle / u_*^2$, and (f) $\langle TKE \rangle / u_*^2$ at $z/H$=0.5 with various integral length scales.

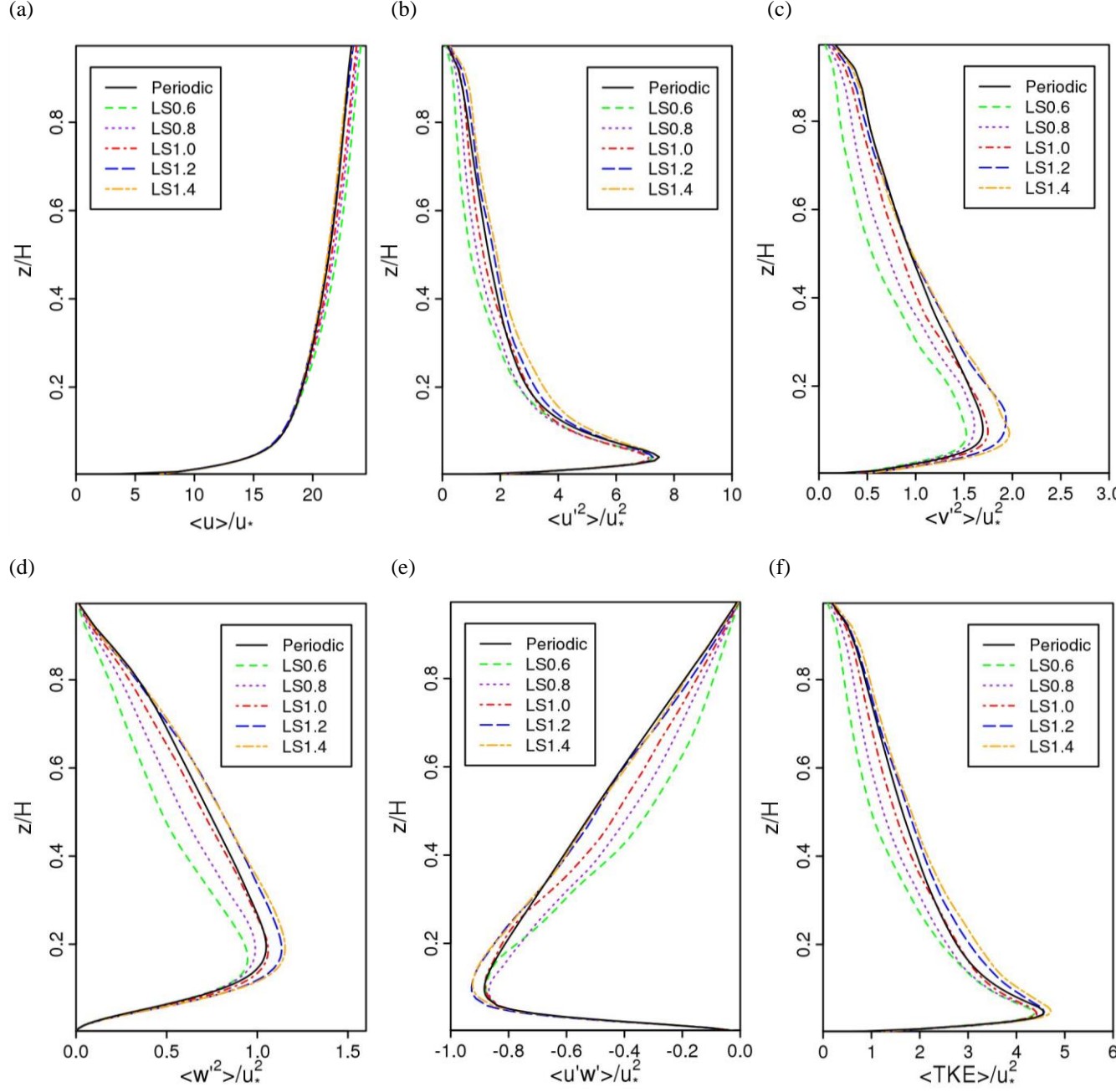

5   Figure 9: Vertical profiles (spatially and temporally averaged) of (a) $\langle u \rangle / u_*$, (b) $\langle u'^2 \rangle / u_*^2$, (c) $\langle v'^2 \rangle / u_*^2$, (d) $\langle w'^2 \rangle / u_*^2$, (e) $\langle u'w' \rangle / u_*^2$, and (f) $\langle TKE \rangle / u_*^2$ at $x/H$=10 with various integral length scales.

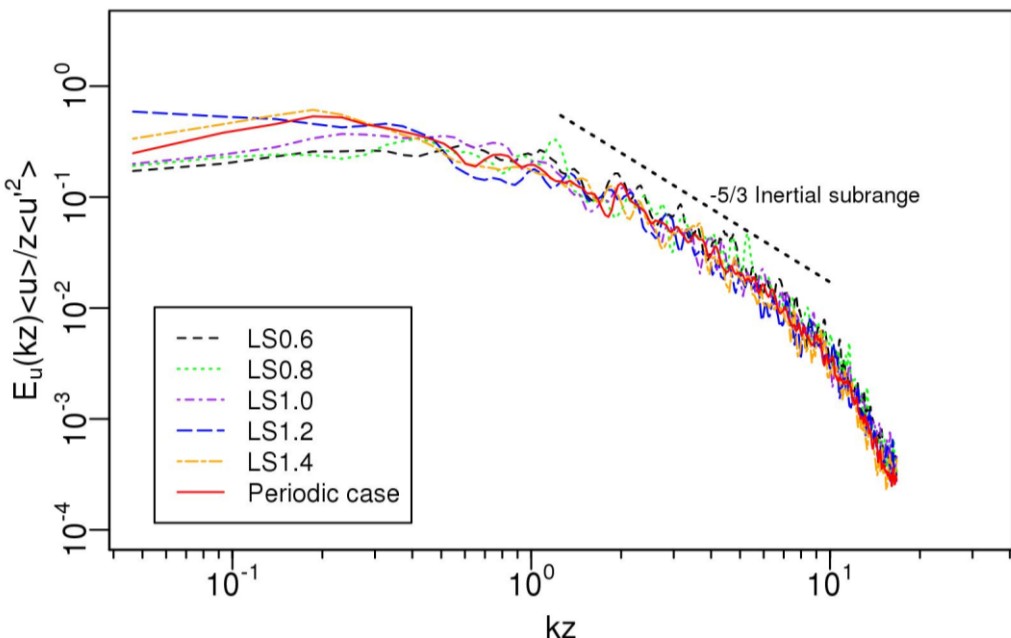

Figure 10: Spectra of streamwise velocity component for a series of downwind locations at *x/H*=10 and *z/H*=0.5 with various integral length scales, *k* is the angular wavenumber, with $\langle u \rangle$ and $\langle u'^2 \rangle$ the spatially averaged mean and streamwise normal turbulent stress, respectively.

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
