# Peer review of "Implementation of a synthetic inflow turbulence generator in idealised WRF v3.6.1 large eddy simulations under neutral atmospheric conditions"

_Geoscientific Model Development, 2019_

## Referee Comment (RC1) · Anonymous Referee #1 · 27 Sep 2019

General comments

This manuscript reports the implementation and testing of the existing synthetic inflow turbulence generation method (Xie and Castro, 2008) for large-eddy simulations (LES). The LES-model in question is the widely used WRF-LES model for small-scale atmospheric problems.

The topic is important since the question of how to deal with the missing inflow-turbulence information is one of the most important issues in the context of practical applications of LES to small-scale atmospheric problems as well as to other kind of turbulent-flow problems such as e.g. many engineering related problems. The authors

correctly point out the particular need for methods to handle the inflow turbulence question in cases when the LES model is nested within a meso-scale atmospheric model domain. The gray zone between the scales resolvable by the meso-scale models and the resolution requirements of LES unavoidably lead to a large gap in the resolution and therefore it becomes very important to somehow incorporate the lacking turbulence information on the inflow boundaries of the LES-domain in some more or less approximative manner.

The degree of novelty of the present work is not particularly high. This is because the method in question was developed already more than ten years ago, and because the same method has already been implemented in some other LES models such as the PALM model which is also a LES model for small-scale atmospheric problems like WRF-LES. However, in my opinion, this work deserves to be published since it involves a rather systematic study of the properties of the method in the WRF-LES model. Especially, the sensitivity study to the integral length scale provides some new and very likely useful information.

A remarkable weakness of the work is that the synthetic-turbulence generation method has not been parallelized. Instead, the root process performs the whole task of the synthetic turbulence generation and then distributes it to those other processes which need this information. As shown in the manuscript (Fig. 1 b), this severely compromises the computational speed even in this kind of rather moderate-sized simulation. In really large simulation set ups with thousands or even tens of thousands of CPU-cores employed, the non-parallelized method becomes totally impractical. Therefore the question about the parallelization must be discussed more deeply in the manuscript. Note that the problem of parallelizing this method has already been solved at least in the PALM-implementation.

Generally, the manuscript is quite well structured and written, but some improvements are needed, see the specific comments below. In part of the figures, especially in Fig. 4, the legend texts are too small, please enlarge them.

[Figure]

Specific comments

Page 2, line 4: I find the statement: "The WRF-LES model can capture the intermittency of three-dimensional turbulent eddies." a bit confusing. This should be clarified.

Page 3, line 16: Just a typo: white noise is typed "while noise".

Page 3, line 27: Perhaps another reference to PALM could be added here, see https://www.geosci-model-dev-discuss.net/gmd-2019-103/ although this is still currently in the discussion phase.

Page 6, lines 5 and 6: "...dominant Reynolds stress tensors..." does this possibly mean dominant Reynolds stress components, or something else? Please correct.

Page 6, lines 14 and 15: "...the vertically same wind direction...". For instance "vertically constant wind direction" would be better wording.

Page 7, lines 21 and 22: The last sentence of this paragraph is obvious and could as well be dropped.

Page 8, lines 8 and 9: "$/u\_*^2$ has a higher value at z/H = 0.1 than that at z/H = 0.5. This is consistent with the trend that it decreases with height in the boundary layer." I find this, too, kind of obvious and unnecessary to mention.

Page 8, lines 14-16: "The slower adjustment...can be attributed to a larger shear-generated TKE..." I don't really understand the line of thinking here. I think this statement should be better justified and explained.

Page 8, Sec. 3.1.4: The inflow case profiles of the second moments in Fig. 5 (and also to some extent in Fig. 9) appear wavy compared with the periodic case profiles. I assume that it is very clear for a large majority of the readers that this is because these profiles are not averaged in the stream-wise direction like those of the periodic case. However, I think this should be nevertheless explained in the text.

Page 9, lines 16 and 17: The last sentence of this paragraph appears vague. Please,

improve it. One reason for my confusion may be that there is no inertial subrange visible in the spectra shown in Fig. 10, probably because of the rather moderate resolution and/or numerically dissipative advection scheme. Moreover, I think that the term "inertial sublayer" is not good. It is better to say inertial subrange because it is not intuitive (or at least I don't find it intuitive) to think about layers in the wave-number space.

Page 9, line 32: "...less 1.0...", please, add "than".

Page 10, line 3: "...the 'accurate' ones...". I assume this refers to that in the case LS1.0 the integral length scales are set as evaluated from the periodic case results, but I am not sure if I understood this correctly. This should be written more clearly.

Page 10, line 3: I guess LE ratio should be LS ratio.

Page 10, line 12, "...WRF-LES (v3.6.1) models...". Why models, i.e. why in pluralis form?

Page 11, lines 5-7: I find these last two sentences of this paragraph very unclear. If this is to discuss the (so far) lacking parallelization of the method, it is not sufficient and not at all clear. As stated above in my general comments this issue must be discussed more deeply. It deserves its own paragraph in the Discussion and conclusions section, but should also be better brought up in Sec. 2.3.
* * *

---

## Referee Comment (RC2) · Anonymous Referee #2 · 23 Nov 2019

The manuscript "Implementation of a synthetic inflow turbulence generator in idealised WRF v3.6.1 large eddy simulations under neutral atmospheric conditions" by Zhong et al. submitted to the Geoscientific Model Development (GMD) describes the implementation of an existing synesthetic turbulence generator to the Weather Research and Forecasting (WRF) model, with the aim of reducing the inflow fetch distance for nested simulations down to the large-eddy simulation (LES) scale. They tested a neutral boundary layer (NBL) case, and performed sensitivity study of a key length scale in their turbulence generator. The results were then evaluated against a standalone periodic LES simulation.

This work will benefit the atmospheric community by providing then with a practical engineering tool for improving nested simulations at the LES scale. Implementing a piece of code like this into WRF is no "a walk in the park", it must have taken the authors a great deal of time and effort. For that I appreciate their efforts, and applaud them for making their code publicly available with this manuscript.

But regarding the contents, I am afraid that I fail to see the scientific novelty with this manuscript. It seems that all they did were to document the performance of an existing method on one particular case. One way to improve this manuscript is for the authors to interpret their results based on more detailed analysis rather than speculation, so that the readers have a more fundamental understanding of the strength and weakness of the synthetic turbulence generator applied to the atmospheric boundary layer flow. For example, regarding Fig. 4f, the authors observed that the TKE profiles at 0.1H requires a longer fetch to converge to the periodic solution, and commented that this maybe due to "downward turbulence transport from above". My suggestion is then don't stop at this speculation, investigate it by plotting the resolved TKE budgets and prove or disapprove your hypothesis. I have listed a few suggestions in the major comments, but the list is by no means exhaustive.

Finally, please, please improve your English writing, proof read it carefully and invite a native speaker to proofread the manuscript before submission. Overall, I suggest major revisions.

Major comments:

1. Add more analysis to help interpret your results, as I have mentioned in the overall comments, speculation is hardly helpful. After you document the various mean profiles and turbulence statistics, analyze them to help us understand why.

2. I suggest the authors add a control case where inflow contains no turbulence information, just the mean profiles. This way the readers could have a much better sense of the advantage/power of the turbulence generator by comparing the results to the
control case.

3. When presenting the various profiles and spectra, I suggest adding profiles/spectra at $x/H = 0$, i.e., the inlet profiles directly from the turbulence generator. This way, we have a better sense of the direct output from turbulence generator.

4. I wonder if the shape of the integral length scale profiles in Fig. 1a matter for the results. The step function like integral length scale in the streamwise direction $L_x$ worries me a little bit, and please elaborate on your "canopy" argument for $L_x$. Furthermore, the relative importance of these integral length scale profiles is also of interest. For example, what if you only vary $L_y$ but keep $L_x$ and $L_z$ the same in your sensitivity tests?

5. The model setup also worries me. In Page 6, your domain depth is 0.5 km, and if I understand correctly based on your Line 7, the boundary layer depth is also 0.5 km. Such a shallow domain depth might cause undesirable reflections back into your domain, unless you are using radiative top boundary conditions. Is that implemented in WRF? Please comment/give more information on the top boundary condition used.

Minor comments:

1. Page 2, Line 4, "The WRF-LES model can capture the intermittency of three-dimensional turbulent eddies". Could you provide a reference please? It would be useful to the readers. I am also curious to learn about studies on turbulence intermittency using WRF-LES.

2. Page 2, Line 5, "There still remains a challenge for downscaling from mesoscale simulation (down to 1 km) to the LES scale (tens of meters or below) (Doubrawa et al., 2018)." Please, summarize brief what this challenge is.

3. Page 2, Line 7, "Most WRF-LES models . . . uses. . ." please fix your grammar.

4. Page 2, Line 8, By "These brave assumptions", you actually meant the one brave assumption of periodic boundary conditions only. Please improve this sentence.

5. Page 2, Line 12. I am confused about your "As one step moving towards enabling WRF's capability of nesting...". Why and how would the synthetic turbulence inflow scheme help with nesting? I guess this is related to my earlier point that you need to lay out clearly the difficulties of meso-to-microscale nesting first, before diving into your proposed method.

6. Page 2, Line 18 "turbulence" not "turbulences".

7. Page 3, Line 21, "It is thus not surprising that a very long distance, e.g. 20 – 40 boundary layer depths, is normally required to allow a transition to fully developed turbulence." This statement might be misleading. My understanding is that the cell perturbation method (CPM) of Munoz-Esparza et al. (2014) applied to potential temperature requires only a short distance before turbulence is properly spun up, even for the neutral boundary layer (see their Fig. 7). This is also true when CPM is applied to velocity (Mazzaro et al., 2019, JAMES, 11(7):2311-2329). The author should clarify or give a proper reference to the fetch distance of "20-40 boundary layer depths".

8. Page 3, Line 25, "flows" not "follows".

9. Page 4, Line 4, "energy-taking resolved eddies" ? This sounds very strange.

10. Eqs. 1-2, perhaps you are using the Favre filter in these equations, or perhaps you are using the Boussinesq approximations for the PBL, please clarify. Eqs.1-2 are not the governing equations for compressible flow as you indicated in Line 3.

11. Eq. 7, this is a parameterized TKE equation where turbulent transport and pressure correlation terms are parameterized. It is also written without the buoyancy term, and should therefore only applicable to a vertical depth within the NBL, but not above the boundary layer where stable stratification prevails. Unless the authors intend to adopt an isentropic background state for their simulations, I suggest including the buoyancy terms for completeness. The use of the mixing length "l" as the dissipation scale is another assumption that should at least be mentioned.

12. Eq. 15, please explain the meaning of the "alpha" inside the matrix. It also looks strange that you shall write a_{ij} in a matrix form in Eq. 15. Shouldn't alpha_{ij} be an element of your matrix, rather the entire matrix itself?

13. Page 6, Line 9, what do you mean by "a constant Lagrangian time scale T (Eq. 13) using Taylor's hypothesis" ? please give more detail here, how did you determine your "constant T" value?

14. Page 6, Line 10-11, "canopy height"? Why suddenly canopy height? What's the purpose of placing a canopy layer in your NBL simulations?

15. Page 6, ". . .,explained in Xie and Castro (2008)". Please fix your grammar.

16. Page 6, Line 14-15, "the vertically same wind direction", please fix your grammar.

17. Page 6, Line 19, "in the lateral direction", did you mean "spanwise" direction? Same for the rest of this paragraph. Lateral suggests both x- and y-directions.

18. Fig. 1, caption, use "relative computation time" as in your main text, rather than "relative computation".

19. Fig. 1, "dashed grey line of 1.0 indicating", indicates, not indicating.

20. Page 7, Line 9, "filtered velocity" rather than "filter velocity".

21. Page 7, Line 14, and elsewhere. Please double-check on the GMD conventions, but I think you should spell out "Figure" if it is at the beginning of a sentence.

22. Fig. 2. Caption, "(b) The . . ." change to "(b) the. . ."

23. Page 7, Line 16, "are advected and decay downwind. . .", please fix your grammar.

24. Page 7, "can generate realistic well-configured turbulence structures from a short adjustment distance downwind". The adjustment distance does not look short to me. Judging from your Fig. 2b, it looks like a fetch distance of x = 5H is required at least. Please comment on this.

25. Page 7, Line 21 to 22, "and there is no adjustment distance, and instead, an adjustment time to generate fully-developed turbulence structures". Please fix your grammar.

26. Page 7, Line 28, "plan" or "plane"?

27. Fig. 3, I suggest using the "global friction velocity ðİŠŰ∗" from the periodic case to normalize u* for the inflow case. This way, we could detect the presence of systematic biases in the inflow case, if any.

28. Fig. 3. caption "(laterally and temporally)", laterally and temporally averaged?

29. Page 8, Line 10, "a good agreement against ?" Please improve this sentence.

30. Page 8, Line 12, can you comment on the possible reason for the slow convergence (long fetch distance ) of w^2 at 0.1 z/H ?

31. Page 8, Line 15, why would "a larger shear-generated TKE" slow down the adjustment at 0.1z/H? Shouldn't this accelerate the adjustment because more TKE is generated locally independent of the TKE contained in the inflow.

32. Page 8, "downward turbulence transport from above" Did you look at the TKE budget? The transport term of TKE is quite insignificant in the NBL. Unless the inflow case is doing something less. It would be nice if you could present the TKE budgets and compare between the two cases.

33. Page 8, "The red circle dots", just "red circles" will do.

34. Page 8, Line 21, "noticed again" or "noted again"?

35. Fig. 6, caption "< ðİŠŰ > and < ðİŠŰ′ > the laterally averaged mean and streamwise normal Reynolds stress", how are these Reynolds stresses? These are first-order moments.

36. Page 9, Line 17, "is able to sustained", please fix your grammar.

37. Fig. 6, could you include a spectrum at the inlet x = 0, so that the readers have an idea of what the synthetic turbulence spectrum looks like?

38. "A length scale (LS) ratio . . . are tested." Please fix your grammar.

39. Page 9, bottom line "Fig. 8 (a) shows that < ðİŚć >/ðİŚć∗ is slightly greater for the LS ratio less 1.0 (see Fig. 8a for comparison). This is due to a greater Reynolds shear stress < ðİŚć′ðİŚď′ >/ðİŚć∗2". I do not understand your explanation. The velocity profile at z/H = 0.5 is affected by the divergence of the stresses, rather than the stress itself. How could a large stress value at z/H = 0.5 explain the overestimation of the velocity?

40. Page 10, Line 1, "Figs. 8(b-d) and (f)" rather than "Fig. 8(b-d)".

41. Page 10, Line 3, what is the "LE ratio"? did you mean your "LS 1.0" case?

42. Page 10, Line 3, why is "LE ratio equal to one" the "accurate ones"? First of all, please fix your grammar. Second, what do you mean by "accurate"?

43. Page 10, Line 5, if all you have to say about Fig. 9 is that it "confirms the findings suggested from Fig. 8", I would suggest you remove that figure.

44. Page 10, Line 9-10, "There is no significant change of the spectra", depends on what you mean by significant. The differences among these LS cases are similar to those presented in Fig. 6. I would suggest you plot your data on kE – log(k) plots. First, this avoids the flat 1D spectra issue at the low wavenumbers. Second, it would be much easier to tell the differences if the y-axis is not on a log scale. 45. Page 10, Line 12, "idealised WRF-LES (v3.6.1) models", model not models

46. Page 11, Line 11, "The spectrum of these data shows an inertial subrange". I strongly recommend you show these in your spectra plots.

47. Page 11, Line 12, "yields a satisfactory accuracy". Please, fix your grammar.

Please also note the supplement to this comment:
https://www.geosci-model-dev-discuss.net/gmd-2019-165/gmd-2019-165-RC2-supplement.pdf

———————————————————————

**[GMDD](https://www.geosci-model-dev-discuss.net)**

---

## Referee Comment (RC3) · Anonymous Referee #3 · 16 Dec 2019

In the manuscript "Implementation of a synthetic inflow turbulence generator in idealised WRF v3.6.1 large eddy simulations under neutral atmospheric conditions" present a methodology for generation of inflow turbulence for large eddy simulations using the Weather Research and Forecasting (WRF) model.

General Remarks

The manuscript attempts to address a timely and relevant problem of inflow turbulence generation in large-eddy simulations of realistic atmospheric boundary layer flows. While there is nothing fundamentally wrong with the methodology applied the manuscript has a number of significant deficiencies. The review of previous work in the

field is inadequate. The authors make several references to derived work instead of citing the original work (more details are provided under "Specific Remarks".) Only neutral boundary layer simulations are carried out and the Coriolis force was not activated. Such setup does not produce a realistic atmospheric boundary layer. Furthermore, the synthetic turbulence generation approach of Xie and Castro (2008) was already implemented in WRF by Muñoz-Esparza et al. (2015), so it is not clear what is the original contribution of this work. Finally, some of the conclusions about the effectiveness of the synthetic turbulence generation approach are not supported by the results presented in the manuscript. In particular, the length of the fetch needed to achieve the equilibrium boundary layer is underestimated.

Taking all the above into account I do not recommend the manuscript for publication in the journal Goescientific Model Development.

Specific Remarks

Page 2, line 2 – The reference to Nottrott et al. is not appropriate, since Nottrott et al. did not develop WRF. Proper reference would be Skamarock and Klemp (JCP 2008).

Page 2, line 7 – Doubrawa et al. 2018 is certainly not the first or most important reference related to WRF-LES.

Page 2, line 11 – This is not an example of a fundamental study. Nunalee et al. (2014) reported on LES using WRF model based on a tracer dispersion field study and compared simulation results to field study observations.

Page 2, line 14 – Munoz-Esparza et al. (PoF 2015) have already implemented synthetic turbulence inflow scheme by Xie and Castro (2008), so it is not clear what is the original contribution of this work.

Page 2, line 20 – A space is missing between year and semicolon, here, and on numerous places throughout the manuscript.

Page 2, line 26 – However, the velocity profile could be modified, also it can vary in

time.

Page 3, line 16 – More recent reference that expands and improves on Muñoz-Esparza et al. (2015) is Muñoz-Esparza and Kosovic (2018).

Page 4, line 4 – A subgrid scale scheme does not parameterize small unresolved eddies, instead it parameterizes the effect of small unresolved eddies on the resolved field.

Page 5, Equation 9, 12, etc. – The notation using plus sign is confusing since subscript m indicates the velocity component.

Page 6, line 15 – Why is Coriolis turned off if simulation of flow in an atmospheric boundary layer is the goal? Page 7, line 6 – Doubling the computational time is a significant increase that needs to be justified.

Page 7, line 7 – The adjustment distance should be more precisely quantified.

Page 8, line 2 – Instead of symbols, the stresses should be defined as: "horizontal profiles of normal and shear turbulent stresses normalized by surface friction velocity."

Page 8, line 7 – Normalized streamwise variance matches well at $x/H = 7$ or 8 and not at $x/H = 5$.

Page 8, line 10 – Below $z/H = 0.3$ the profile of cross-stream variance differs significantly for any $x/H$.

Page 8, line 12 – The development is not achieved at all, since only at the end of the domain the values of $<w'2>/u*2$ obtained using the synthetic turbulence generation method are the same as those from the simulation involving periodic domain. Also, what is shown in the figures is the fetch, not the time scale.

Page 8, line 13 – Figures show that the fetch needed for different quantities to reach the equilibrium values differs significantly between them. For example, vertical velocity variance does not reach equilibrium. Since it is a component of TKE, TKE also requires

a long fetch to reach the equilibrium.

Page 8, line 18 – Same as above, these should be labeled as normal and shear turbulent stresses normalized by surface friction velocity.

Page 8, line 22 – A sentence should not begin with a symbol.

Page 8, line 23 – In "matches closely to that...," "to" should be omitted.

Page 8, line 24 – Same as above, instead of symbols names of the terms should be used.

Page 9, line 9 - The spectral roll-off depends on the numerics not on the turbulence generation scheme, so this is questionable conclusion. Also, flat spectra over a decade of wave numbers is not realistic. Furthermore, there is not apparent inertial range (-5/3) slope in the results presented in Figure 6.

Page 9, line 14 – If current work does not differ from Munoz-Esparaza et al. (2015), what is new in the present manuscript?

Page 9, line 24 – Instead of "slightly affects," it should be "affects slightly."

Page 9, line 30 – As before, words should be used instead of symbols.

Page 10, line 3 – It is not clear what is meant by " 'accurate' ones..."

Page 10, line 16 – It is not clear what is the purpose of the statement starting with "It is not trivial..." This statement by itself is of little relevance, the question is: What is the relevance?

Page 10, line 21 – The adjustment fetch should be quantified. It is not short.

Page 11, line 12 – The statement related to "...a satisfactory accuracy" is an imprecise qualitative statement. It should be stated what is the accuracy satisfactory in comparison to.

---

## Author Comment (AC1) · 15 Feb 2020

**Responses to the comments from Anonymous Referee #1**

**General comments**

**Comment G1:** *This manuscript reports the implementation and testing of the existing synthetic inflow turbulence generation method (Xie and Castro, 2008) for large-eddy simulations (LES). The LES model in question is the widely used WRF-LES model for small-scale atmospheric problems. The topic is important since the question of how to deal with the missing inflow turbulence information is one of the most important issues in the context of practical applications of LES to small-scale atmospheric problems as well as to other kind of turbulent-flow problems such as e.g. many engineering related problems. The authors correctly point out the particular need for methods to handle the inflow turbulence question in cases when the LES model is nested within a meso-scale atmospheric model domain. The gray zone between the scales resolvable by the meso-scale models and the resolution requirements of LES unavoidably lead to a large gap in the resolution and therefore it becomes very important to somehow incorporate the lacking turbulence information on the inflow boundaries of the LES-domain in some more or less approximative manner.*

*The degree of novelty of the present work is not particularly high. This is because the method in question was developed already more than ten years ago, and because the same method has already been implemented in some other LES models such as the PALM model which is also a LES model for small-scale atmospheric problems like WRF-LES. However, in my opinion, this work deserves to be published since it involves a rather systematic study of the properties of the method in the WRF-LES model. Especially, the sensitivity study to the integral length scale provides some new and very likely useful information.*

**Response:** We thank the reviewer for the overall positive comments. To echo the positive feedback from the reviewer, we wish to reiterate that the gray zone issue still remains challenging for sub-kilometre meteorological modelling and there is a great demand for a reliable nesting methodology to enable sub-hundred-metre large-eddy simulations of the atmospheric boundary layer. The WRF model is perhaps the best platform to test such a methodology, whilst PALM has no capability of meso-scale meteorological modelling. As one important step to achieve this target, this study attempts to equip WRF with a well-tested synthetic turbulence inflow scheme (Xie and Castro 2008), which has been implemented and tested on engineering type of codes, such as Star-CD (Xie and Castro, 2009) and OpenFOAM (Kim and Xie, 2016) and the micro-scale meteorology code PALM (PALM, 2017; Maronga et al., 2019). We believe that this new capability will benefit many boundary layer modellers to run a LES for their local areas nested online within a meso-scale domain.

The focus of this paper is to rigorously test and explore the Xie & Castro (2008) method in the meso-to-micro-scale meteorological code WRF, in terms of the sensitivity of integral length scales and the adjustment distance of the mean velocity field, the turbulent Reynolds stresses and the local friction velocity. These are the novelties of the paper.

**Comment G2:** *A remarkable weakness of the work is that the synthetic-turbulence generation method has not been parallelised. Instead, the root process performs the whole task of the synthetic turbulence generation and then distributes it to those other processes which need this information. As shown in the manuscript (Fig. 1 b), this severely compromises the computational speed even in this kind of rather moderate-sized simulation. In really large simulation set ups with thousands or even tens of thousands of CPU-cores employed, the non-parallelised method becomes totally impractical. Therefore the question about the parallelisation must be discussed more deeply in the manuscript. Note that the problem of parallelising this method has already been solved at least in the PALM-implementation.*

**Response:** This study is focused on the feasibility of implementing the inflow method (Xie & Castro, 2008) in the meso-to-micro-scale meteorological code WRF and the impact of the key variables (i.e. the integral length scales) on the simulated turbulence development inside the domain. Up to the authors' knowledge, the latter has not been rigorously addressed previously. We appreciate that *the technical parallelisation of the Xie & Castro (2008) method has been done in PALM and that some other researchers (e.g. Kim and Xie, 2016) have also made efforts to technically parallelise the Xie & Castro method. These suggest that technically parallelising this method is not an issue. It is our intention that we test the method inside WRF scientifically and rigorously here and publish our open source code through GMD to allow other WRF-LES users to extend technical capabilities of our code, such as parallelisation.* In response to the reviewer's comment, the following paragraph has been added in Discussion and conclusions:

"This study is focused on the feasibility of implementing the inflow method (Xie & Castro, 2008) in the meso-to-micro-scale meteorological code WRF and the impact of the key variables (i.e. the integral length scales) on the simulated turbulence development inside the domain. This inflow subroutine has previously been implemented in both serial and parallel mode in several codes, including engineering type of codes Star-CD (Xie and Castro, 2009) and OpenFOAM (Kim and Xie, 2016), and the micro-scale meteorology code PALM (PALM, 2017). Although the current implementation in WRF is affordable for a moderate-sized simulation (e.g. tens of meters resolutions), the technical parallelisation of this inflow subroutine in WRF-LES can be the future work for very large simulation domains with high resolutions. Users of our open source subroutine may offer this technical contribution."

**Comment G3:** *Generally, the manuscript is quite well structured and written, but some improvements are needed, see the specific comments below. In part of the figures, especially in Fig. 4, the legend texts are too small, please enlarge them.*

**Response:** The specific comments are responded below. The legend texts in Figs 1, 4, 5, 8 and 9 are enlarged.

**Specific comments**

**Comment:** *Page 2, line 4: I find the statement: "The WRF-LES model can capture the intermittency*

*of three-dimensional turbulent eddies." a bit confusing. This should be clarified.*

**Response:** This sentence is deleted and has been replaced by a statement attached to the previous sentence: "At the microscale, a large eddy simulation (LES) can be activated in the WRF model (WRF-LES), enabling users to simulate the characteristics of energy-containing eddies in the atmospheric boundary layer.".

**Comment:** *Page 3, line 16: Just a typo: white noise is typed "while noise".*

**Response:** This is corrected.

**Comment:** *Page 3, line 27: Perhaps another reference to PALM could be added here, see*

*https://www.geosci-model-dev-discuss.net/gmd-2019-103/ although this is still currently in the discussion phase.*

**Response:** This additional reference for PALM has been added.

**Comment:** *Page 6, lines 5 and 6: "...dominant Reynolds stress tensors..." does this possibly mean dominant Reynolds stress components, or something else? Please correct.*

**Response:** "dominant Reynolds stress tensors" is replaced with "Reynolds stress components".

**Comment:** *Page 6, lines 14 and 15: "...the vertically same wind direction...". For instance "vertically constant wind direction" would be better wording.*

**Response:** "the vertically same wind direction" is replaced with "the vertically constant wind direction".

**Comment:** *Page 7, lines 21 and 22: The last sentence of this paragraph is obvious and could as well be dropped.*

**Response:** This sentence is removed now.

**Comment:** *Page 8, lines 8 and 9: "$/u\_*^2$ has a higher value at z/H = 0.1 than that at z/H = 0.5. This is consistent with the trend that it decreases with height in the boundary layer." I find this, too, kind of obvious and unnecessary to mention.*

**Response:** This is removed now.

**Comment:** *Page 8, lines 14-16: "The slower adjustment...can be attributed to a larger shear generated TKE..." I don't really understand the line of thinking here. I think this statement should be better justified and explained.*

**Response:** This sentence is removed. We have added the following discussion regarding the developing distance for TKE in Section 3.1.3.

"Since the streamwise velocity variance has a major contribution to TKE, the developing distance for TKE is similar to that for the streamwise velocity variance, i.e. about $x/H = 7$-8. The distance needed for different quantities to develop the turbulence differs between each other, and it is about $x/H = 5$-15."

**Comment:** *Page 8, Sec. 3.1.4: The inflow case profiles of the second moments in Fig. 5 (and also to some extent in Fig. 9) appear wavy compared with the periodic case profiles. I assume that it is very clear for a large majority of the readers that this is because these profiles are not averaged in the stream-wise direction like those of the periodic case. However, I think this should be nevertheless explained in the text.*

**Response:** We have reprocessed the model output with much smaller time intervals (5 sec now compared with 60 sec previously). The revised profiles in Figs. 5 and 9 are now much smoother.

**Comment:** *Page 9, lines 16 and 17: The last sentence of this paragraph appears vague. Please, improve it. One reason for my confusion may be that there is no inertial subrange visible in the spectra shown in Fig. 10, probably because of the rather moderate resolution and/or numerically dissipative advection scheme. Moreover, I think that the term "inertial sublayer" is not good. It is better to say inertial subrange because it is not intuitive (or at least I don't find it intuitive) to think about layers in the wave-number space.*

**Response:** In the previous version, the spectra were calculated based on the spatially distributed data along the cross-stream direction ($y$) for given ($x$, $z$) coordinates. These were averaged for a number of time steps to smooth them. The limit of this approach is that a small number of data along the cross-stream direction ($y$) were used.

A slightly different approach is adopted in the revised paper. For given ($x$, $y$, $z$) coordinates, a spectrum was calculated based on a time series of 3600 s with an interval of 5 s. Five spectra for ($y/H = 1.76, 2.16, 2.56, 2.96 \ and \ 3.36$) at the same ($x$, $z$) coordinates were averaged to obtain a smoother one plotted in Fig. 6. The same was used for the new Figure 10. In the text, "inertial sublayer" is replaced by "inertial subrange".

We also added some discussion as,

"It is noted that for a very high resolution, e.g. in the order of magnitude 1 meter, similar as that used in the simulations of PALM (PALM, 2017), the inertial subrange in the spectrum is wider."

**Comment:** *Page 9, line 32: "...less 1.0...", please, add "than".*

**Response:** "than" is added.

**Comment:** *Page 10, line 3: "...the 'accurate' ones...". I assume this refers to that in the case LS1.0 the integral length scales are set as evaluated from the periodic case results, but I am not sure if I understood this correctly. This should be written more clearly.*

**Response:** "the 'accurate' ones" is replaced with "the 'accurate' (compared with the periodic case) one".

**Comment:** *Page 10, line 3: I guess LE ratio should be LS ratio.*

**Response:** "LE ratio equal to one" is replaced with "the LS 1.0 case".

**Comment:** *Page 10, line 12, "...WRF-LES (v3.6.1) models...". Why models, i.e. why in pluralis form?*

**Response:** "idealised WRF-LES (v3.6.1) models" is replaced with "an idealised WRF-LES (v3.6.1) model".

**Comment:** *Page 11, lines 5-7: I find these last two sentences of this paragraph very unclear. If this is to discuss the (so far) lacking parallelisation of the method, it is not sufficient and not at all clear. As stated above in my general comments this issue must be discussed more deeply. It deserves its own paragraph in the Discussion and conclusions section, but should also be better brought up in Sec. 2.3.*

**Response:** See the response to Comment G2. We have added a paragraph for the discussion about the parallelisation of the method in the third paragraph of the section of Discussion and conclusions

---

## Author Comment (AC2) · 15 Feb 2020

**Responses to the comments from Anonymous Referee #2**

**General comments**

**Comment G1:** *The manuscript "Implementation of a synthetic inflow turbulence generator in idealised WRF v3.6.1 large eddy simulations under neutral atmospheric conditions" by Zhong et al. submitted to the Geoscientific Model Development (GMD) describes the implementation of an existing synesthetic turbulence generator to the Weather Research and Forecasting (WRF) model, with the aim of reducing the inflow fetch distance for nested simulations down to the large-eddy simulation (LES) scale. They tested a neutral boundary layer (NBL) case, and performed sensitivity study of a key length scale in their turbulence generator. The results were then evaluated against a standalone periodic LES simulation.*

*This work will benefit the atmospheric community by providing then with a practical engineering tool for improving nested simulations at the LES scale. Implementing a piece of code like this into WRF is no "a walk in the park", it must have taken the authors a great deal of time and effort. For that I appreciate their efforts, and applaud them for making their code publicly available with this manuscript.*

**Response:** We appreciate the reviewer's comments on the challenges on implementing an inflow synesthetic turbulence generator in the WRF-LES model. The inflow method was originally developed for engineering applications, and has not been rigorously tested in full-scale atmospheric boundary layer problems. This study extended a well-tested synthetic turbulence inflow scheme (Xie and Castro 2008) into the WRF-LES model. This implementation can be applied to the WRF-LES simulation with a multi-scale seamless nesting case from a meso-scale domain with a km-resolution (where the time-averaged information is known, which can be used as the inputs for the synthetic inflow turbulence generator) down to LES domains with metre resolutions (with additional turbulent information).

**Comment G2:** *But regarding the contents, I am afraid that I fail to see the scientific novelty with this manuscript. It seems that all they did were to document the performance of an existing method on one particular case. One way to improve this manuscript is for the authors to interpret their results based on more detailed analysis rather than speculation, so that the readers have a more fundamental understanding of the strength and weakness of the synthetic turbulence generator applied to the atmospheric boundary layer flow. For example, regarding Fig. 4f, the authors observed that the TKE profiles at 0.1H requires a longer fetch to converge to the periodic solution, and commented that this maybe due to "downward turbulence transport from above". My suggestion is then don't stop at this speculation, investigate it by plotting the resolved TKE budgets and prove or disapprove your hypothesis. I have listed a few suggestions in the major comments, but the list is by no means exhaustive.*

**Response:** We implemented a synthetic turbulence inflow generator (Xie and Castro 2008), which has been implemented and tested on engineering type of codes, such as Star-CD (Xie and Castro, 2009) and OpenFOAM (Kim and Xie, 2016) and the micro-scale meteorology code PALM (PALM, 2017; Maronga et al., 2019), into the WRF-LES model. The focus of this paper is to rigorously test and explore the Xie & Castro (2008) method in a full scale (i.e. very large Re number), in terms of the sensitivity of integral length scales and the adjustment distance of the mean velocity field, the turbulent Reynolds stresses, TKE and the local friction velocity. Our paper will be useful to the users of the Xie & Castro (2008) method implemented in meso-scale models, such as WRF, and the micro-scale meteorology models, such as PALM. Our conclusion in the current paper is that the Xie and Castro (2008) method needs 5-15 boundary layer depths to fully develop the turbulence, and this is consistent with those in Xie & Castro (2008), Kim et al (2013) for engineering scale problems. For a coarser grid resolution of 90 m (vs 20 m in our paper), Munoz-Esparza et al. (2014, 2015) tested both their proposed 'cell perturbation method' and the Xie and Castro (2008) method; they concluded that *the cell perturbation*

*method needs a fetch of 15-40 boundary layer depths to fully develop the turbulence, while the Xie and Castro (2008) method needs a longer fetch.* A significant improvement of this fetch generated by our code is one of the novelties and, together with the study of the impact of the key variables (i.e. the integral length scales) on the simulated turbulence development represents the scientific novelties of the paper.

In response to the reviewer's suggestion to improve the interpretation of the results, we have conducted more detailed analyses (see those responses to each individual comment below, and the revised figures in the paper).

With regards to the comment on discussions of Fig. 4f, we have reprocessed the model output with much smaller time intervals (5 sec now compared with 60 sec previously). The revised profiles in Figs. 5 and 9 are now much smoother. Our statement in the previous version does not stand anymore. Therefore we have removed those sentences. Subsequently, we think it is not necessary to look into the TKE budget. The modified text (in Section 3.1.3) is as below:

 "Since the streamwise velocity variance has a major contribution to TKE, the developing distance for TKE is similar to that for the streamwise velocity variance, i.e. about $x/H = 7$-8".

**Comment G3:** *Finally, please, please improve your English writing, proof read it carefully and invite a native speaker to proofread the manuscript before submission. Overall, I suggest major revisions.*

**Response:** We checked our English writing, proofed read the revised manuscript carefully, and also invited a native speaker to proofread the manuscript before re-submission.

**Major comments**

**Comment M1:** *1. Add more analysis to help interpret your results, as I have mentioned in the overall comments, speculation is hardly helpful. After you document the various mean profiles and turbulence statistics, analyze them to help us understand why.*

**Response:** We thank the reviewer for this comment. We have re-postprocessed the model output. In particular, we have now used a much big dataset to generate better statistical results and velocity spectra. These have largely helped us to make more solid conclusions rather than speculations.

In response to the reviewer's comment (also please see our reply to General Comment G2), we have conducted the following extra analyses and added interpretations of the results. Correspondingly, the following text (on Paragraph 2 of Section 2.3) has been added or modified in the manuscript:

"The further 1 h outputs with 5 second interval (~ the advection timescale of the smallest resolved eddies, which is equivalently twice the grid resolution of 20 m) were used for the analysis. In this study, by taking advantage of the homogeneous turbulence in the spanwise direction (Ghannam et al., 2015), we calculate all resolved-scale turbulent quantities by averaging in the spanwise (the *y*-direction) direction and in time *t* over the last 1 h period. This averaging is referred to as "the *y-t* averaging" hereafter, and is denoted by $\langle\varphi\rangle$, for example, for the *y-t* averaged $\varphi$. For a 4D variable, $\varphi(t,x,y,z)$, the *y-t* averaged $\varphi$ is a function of $x,z$, i.e. $\langle\varphi\rangle(x,z)$; for a variable defined on the *x-y* plane, e.g. friction velocity $u_*(t,x,y)$, the *y-t* averaging $u_*$ is a function of *x*, i.e. $\langle u_*\rangle(x)$."

In this way, a better representation of resolved turbulent statistics is achieved. The various curves in the plots are smoother for clearer interpretations. The spectra cover the information of a wider range of eddy sizes.

We have modified and added the following for the explanation of the new spectrum plots (Section 3.1.5):

"For each $x$-location, e.g. $x/H = 10$, the spectrum for the inflow case was firstly calculated from the streamwise wind velocity component over a time series of 3600 s with an interval of 5 s for five selected sample locations of $y_n$ ($y/H = 1.76, 2.16, 2.56, 2.96$ $and$ $3.36$), namely, $\tilde{u}(t, 2H, y_n, 0.5H)$. The spectral data were then averaged over $y_n$ to give the spectra plotted in Fig. 6. "

Another new case with mean inflow only containing no inlet velocity perturbations has been conducted. The horizontal slice of instantaneous streamwise velocity component had been added into Fig 2 as a comparison, in order to provide a better understanding of the advantage of this synthetic turbulence generator.

The spatially and temporally averaged vertical profiles of the mean velocity and the Reynolds stresses, and spectrum for *x/H*=0 for the inflow cases have now been added in the corresponding figures. These provide a better understanding of the direct output from the inflow turbulence generator.

More discussion for the interpreting the results are added, see the following responses.

**Comment M2:** *2. I suggest the authors add a control case where inflow contains no turbulence information, just the mean profiles. This way the readers could have a much better sense of the advantage/power of the turbulence generator by comparing the results to the control case.*

**Response:** We have run one further case with mean inflow only containing no inlet velocity perturbations. The horizontal slice of instantaneous streamwise velocity components had been added into Fig. 2 as a comparison, in order to provide evidence of the advantage of this synthetic turbulence generator. There is nearly no turbulence generated in the domain even after several hours of simulation (also indicated by the following plot for the vertical profile of TKE - note all of the data, except for the Periodic case, are zero). We have added the following discussions in the revised paper (Paragraph 1 in Section 3.1.1):

"For the inflow case without inlet velocity perturbations, there is nearly no turbulence generated in the domain even after several hours of simulation. This is consistent with other similar tests using engineering CFD codes with no synthetic turbulence added at the inlet, e.g. (Xie and Castro, 2008), which demonstrated that a very long distance (e.g 100 times boundary layer thickness) is needed to allow turbulence to develop. This indicates the importance of imposing synthetic turbulence, or at least some form of random perturbations (e.g. Munoz-Esparza et al., 2015) at the inlet. The inflow case without inlet velocity perturbations is not presented in the later sections. "

[Figure]

**Figure R1: Vertical profile of TKE for the inflow case without inlet velocity perturbations, and for the periodic case.**

**Comment M3:** *3. When presenting the various profiles and spectra, I suggest adding profiles/spectra at x/H = 0, i.e., the inlet profiles directly from the turbulence generator. This way, we have a better sense of the direct output from turbulence generator.*

**Response:** The *x/H=0* profiles for the inflow cases are now added into Figs 5 and 6. The turbulence statistics  derived from the current periodic case are used as the input of the inflow turbulence generator. The following are added:
"It is noted that the profiles of the mean velocity and second order moments at the inlet ($x/H = 0$) are overall in a good agreement with these of the periodic case, which further suggests a satisfactory performance of the turbulence generator." (in Paragraph 1 of Section 3.1.4)

"It is shown that the spectrum at the inlet (*x/H*=0) possesses the most broad range of the -5/3 slope compared to the others. There is an evidence of the tendency in the profiles from the inlet downstream to recover to that of the periodic case. The spectrum drops slightly at high wavenumbers from the imposed spectra at $x/H = 0$ to downwind locations, and to recover towards the spectrum of the periodic case. The slight drop suggests a decay of small eddies due to the SGS and molecular viscosities." (in Paragraph 2 of Section 3.1.5)

**Comment M4:** *4. I wonder if the shape of the integral length scale profiles in Fig. 1a matter for the results. The step function like integral length scale in the streamwise direction Lx worries me a little bit, and please elaborate on your "canopy" argument for Lx. Furthermore, the relative importance of these integral length scale profiles is also of interest. For example, what if you only vary Ly but keep Lx and Lz the same in your sensitivity tests?*

**Response:** Xie and Castro (2008) and Kim et al (2013) have already reported more sensitivity studies on the effect of integral length scales, including keeping *Lx* the same and varying *Ly* and *Lz*. They found that a 50% variation in *Ly* and *Lz* generated a variation less than 4% in the friction velocity, and suggested that for the integral lengths not too far from realistic ones, the turbulent statistics are not very sensitive to the length scales.

Again, we emphasise that the aim here is not to generate a particularly accurate simulation of turbulent atmospheric boundary layer flow. Rather, our intention is to assess the adequacy and potential of the inflow generation technique for the prediction of up to second order moments of turbulent statistics.

It is difficult to analyse mathematically the effect of the step change of integral length scales.  However, practically we have not noticed an evident issue. These are consistent with Veloudis et al (2007) and Xie and Castro (2008).

Veloudis, I., Yang, Z., McGuirK, J.J., Page, G.J., Spencer, A.: Novel implementation and assessment of a digtial filter based approach for the generation of LES inlet conditions. Flow Turbul. Combust. 79(1), 1–24 (2007)

We have added the modified text on Paragraph 1 of Section 2.3:
"The streamwise length scale ($L_x$) is specified based on the mean streamwise velocity profile ($\langle u \rangle$) and a constant Lagrangian time scale $T$ (prescribed in Eq. 13), i.e.  $L_x = T\langle u \rangle$ using Taylor's hypothesis (turbulence is assumed to be frozen while it is moving downstream with a mean speed of $\langle u \rangle$). The spanwise length scale ($L_y$) is specified a constant value. The vertical length scale ($L_z$) is specified a smaller constant value near the bottom and a larger constant value for the upper domain to be closer to the measured length scales, as explained in Xie and Castro (2008) and Veloudis et al. (2007). We conducted  a sensitivity study of integral length scales by varying all three baseline $L_x, L_y$ and $L_z$ with a same ratio of 0.6, 0.8, 1.0, 1.2, or 1.4; these individual cases are denoted by "LS0.6", "LS0.8", "LS1.0", "LS1.2", "LS1.4", respectively, in which "LS1.0" is the base  case."

**Comment M5:** *5. The model setup also worries me. In Page 6, your domain depth is 0.5 km, and if I understand correctly based on your Line 7, the boundary layer depth is also 0.5 km. Such a shallow domain depth might cause undesirable reflections back into your domain, unless you are using radiative top boundary conditions. Is that implemented in WRF? Please comment/give more information on the top boundary condition used.*

**Response:** For the neutral boundary layer, the results at any altitudes scaled by the boundary layer height could be interpreted for and applied to the cases with other boundary layer heights, e.g. 1000 m.

We has added the following information about the top boundary conditions used in this WRF-LES model, to respond to the reviewer's comment:
"At the top boundary, a rigid lid (top_lid in the namelist.input file of the WRF-LES model) is specified, and a Rayleigh damping layer of 50 m is used to prevent undesirable reflections (Nottrott et al., 2014; Ma and Liu, 2017) and to maintain a neutral atmospheric boundary layer."

**Minor comments**

**Comment:** *1. Page 2, Line 4, "The WRF-LES model can capture the intermittency of three dimensional turbulent eddies". Could you provide a reference please? It would be useful to the readers. I am also curious to learn about studies on turbulence intermittency using WRF-LES.*

**Response:** This sentence is deleted and has been replaced by a statement attached to the previous sentence: "At the microscale, a large eddy simulation (LES) can be activated in the WRF model (WRF-LES), enabling users to simulate the characteristics of energy-containing eddies in the atmospheric boundary layer."

**Comment:** *2. Page 2, Line 5, "There still remains a challenge for downscaling from mesoscale simulation (down to 1 km) to the LES scale (tens of meters or below) (Doubrawa et al., 2018)." Please, summarize brief what this challenge is.*

**Response:** More details are added on Paragraph 1 Section 1:

"There still remains a challenge for downscaling from a mesoscale simulation (resolutions down to 1 km, capturing mean information only) to an LES scale (tens of meters or below, capturing additional turbulence information) (Doubrawa et al., 2018; Talbot et al., 2012; Chu et al., 2014; Liu et al., 2011), e.g. the appropriate inflow conditions for an LES domain, and the sub-grid scale turbulence schemes suitable for appropriate treatment of the "gray-zone" resolution domain where neither planetary boundary layer (PBL) nor LES parametrisation schemes apply well."

**Comment:** *3. Page 2, Line 7, "Most WRF-LES models : : : uses: : :" please fix your grammar.*

**Response:** "uses" is replaced with "use".

**Comment:** *4. Page 2, Line 8, By "These brave assumptions", you actually meant the one brave assumption of periodic boundary conditions only. Please improve this sentence.*

**Response:** This sentence is improved as follows:

"However, implicit in the use of periodic boundary conditions is the assumption that atmospheric fields and the underlying landuse have repeated periodic features. This assumption may be unrealistic for real landscapes where landuse patterns - and the atmospheric phenomena coupled to them - can be very heterogeneous."

**Comment:** *5. Page 2, Line 12. I am confused about your "As one step moving towards enabling WRF's capability of nesting: : :". Why and how would the synthetic turbulence inflow scheme help with nesting? I guess this is related to my earlier point that you need to lay out clearly the difficulties of meso-to-microscale nesting first, before diving into your proposed method.*

**Response:** As mentioned in a response above, there are two key challenges: appropriate sub-scale turbulence schemes and suitable inflow conditions. In this study, we are focusing on the latter, as one step moving forward. Without the synthetic turbulence inflow scheme, it would take a large distance in the LES domain for the simulated turbulent fields to fully develop. The modified text is:

"Here we implement a well-tested synthetic turbulence inflow scheme (Xie and Castro 2008) in the WRF-LES model (v.3.6.1), in which the meso-scale model could provide the mean flow information as the input of the synthetic turbulence inflow scheme. This scheme provides a step towards enabling WRF's capability of nesting micro-scale turbulent flows within realistic meso-scale meteorological fields."

**Comment:** *6. Page 2, Line 18 "turbulence" not "turbulences".*

**Response:** This is corrected.

**Comment:** *7. Page 3, Line 21, "It is thus not surprising that a very long distance, e.g. 20 –40 boundary layer depths, is normally required to allow a transition to fully developed turbulence." This statement might be misleading. My understanding is that the cell perturbation method (CPM) of Munoz-Esparza et al. (2014) applied to potential temperature requires only a short distance before turbulence is properly spun up, even for the neutral boundary layer (see their Fig. 7). This is also true when CPM is applied to velocity (Mazzaro et al., 2019, JAMES, 11(7):2311-2329). The author should clarify or give a proper reference to the fetch distance of "20-40 boundary layer depths".*

**Response:** We thank the reviewer for pointing this out. Figure 7 in Munoz-Esparza et al. (2014) is just a contour plot without any quantitative information. In their later paper using the Cell Perturbation Method (CPM) for neutral boundary layer, Mazzaro et al., 2019 concludes that "*while the CPM significantly reduced the effect of these high-TKE regions, with a shorter fetch of 15–20 km*" (See their conclusion), which is expected to be consistent with the Fig. 7 in Munoz-Esparza et al. (2014). The neutral boundary layer height used in their papers is 500 m, and a fetch of 15–20 km is equivalent to 30 –40 boundary layer depths.  Also, in another paper Munoz-Esparza et al. (2015), Fig. 10 shows a quantitative profile of Reynolds-shear stress and the resolved TKE for the development distance, in which a fetch of 15-40 boundary layer depths is mentioned for the turbulence development, while 15 boundary layer depths can achieve values within 10% of the quasi-equilibrium solution for cell perturbation method.

It is to be noted that in Munoz-Esparza et al. ( 2014, 2015) and (Mazzaro et al., 2019), the inflow forcing is implemented at the west and south boundaries (i.e. both $x$- and $y$-directions), while we implemented the inflow turbulence generation at the $x$=0 boundary only.  Again, we agree with the authors that the

Cell Perturbation Method (CPM) provides an alternative way of turbulence generation in the modelling of atmospheric boundary layer.

We have added the reference and modified this sentence (on Paragraph 3 of Section 1):

"It is thus not surprising that a large distance of about 20-40 boundary layer depths (Munoz-Esparza et al., 2015; Mazzaro et al., 2019) is normally required to allow a transition to fully developed turbulence."

**Comment:** *8. Page 3, Line 25, "flows" not "follows".*

**Response:** "follows" is replaced with "flows".

**Comment:** *9. Page 4, Line 4, "energy-taking resolved eddies" ? This sounds very strange.*

**Response:** "large energy-taking resolved eddies" is replaced with "large energy-containing eddies at the resolved scale".

**Comment:** *10. Eqs. 1-2, perhaps you are using the Favre filter in these equations, or perhaps you are using the Boussinesq approximations for the PBL, please clarify. Eqs.1-2 are not the governing equations for compressible flow as you indicated in Line 3.*

**Response:** The WRF-LES solves the fully compressible equations in the flux form which implies an application of the Favre filter, formulated using a terrain-following hydrostatic-pressure vertical coordinate. For an LES domain with flat terrain, the momentum equations can be presented by Equation (2). With an assumption of incompressibility of the atmospheric boundary layer, the continuity equation can be expressed as Equation (1). Being rigorous, we change Equation (1) to the original format by removing the assumption of incompressibility. These are also adopted by other WRF-LES studies (Nottrott et al., 2014; Munoz-Esparza et al., 2015).

**Comment:** *11. Eq. 7, this is a parameterized TKE equation where turbulent transport and pressure correlation terms are parameterized. It is also written without the buoyancy term, and should therefore only applicable to a vertical depth within the NBL, but not above the boundary layer where stable stratification prevails. Unless the authors intend to adopt an isentropic background state for their simulations, I suggest including the buoyancy terms for completeness. The use of the mixing length "l" as the dissipation scale is another assumption that should at least be mentioned.*

**Response:** In response to the comment, we have added the buoyancy term in the equation. Since this study is focused on the inflow turbulence generator using WRF-LES in which the subscale TKE equation is coded based on parameterised terms, we consider it appropriate to present the equation in the parameterised forms.

We have added "dissipation coefficient (for more details about the parameterisation see Moeng et al. (2007))."

**Comment:** *12. Eq. 15, please explain the meaning of the "alpha" inside the matrix. It also looks strange that you shall write a_{ij} in a matrix form in Eq. 15. Shouldn't alpha_{ij} be an element of your matrix, rather the entire matrix itself?*

**Response:** To avoid any misunderstanding, $\alpha_{ij}$ is changed to $[\alpha_{i\beta}]$ to represent the matrix form, while $\alpha_{i\beta}$ in the matrix represents an element of the matrix, following the notations of Equation (18) in Xie and Castro (2008). The calculations of $\alpha_{i\beta}$ follow an iterative order: $\alpha_{11}$, $\alpha_{21}$, $\alpha_{22}$, $\alpha_{31}$, $\alpha_{32}$, and $\alpha_{33}$. This has been added in the manuscript.

**Comment:** *13. Page 6, Line 9, what do you mean by "a constant Lagrangian time scale T (Eq. 13) using Taylor's hypothesis" ? please give more detail here, how did you determine your "constant T" value?*

**Response:** This is explained in more details on Paragraph 1 of Section 2.3:

"The streamwise length scale $(L_x)$ is specified based on the mean streamwise velocity profile $(\langle u \rangle)$ and a constant Lagrangian time scale $T$ (prescribed in Eq. 13), i.e. $L_x = T\langle u \rangle$ using Taylor's hypothesis (turbulence is assumed to be frozen while it is moving downstream with a mean speed of $\langle u \rangle$)."

**Comment:** *14. Page 6, Line 10-11, "canopy height"? Why suddenly canopy height? What's the purpose of placing a canopy layer in your NBL simulations?*

**Response:** These have been removed as they are not very relevant to this paper. The modified text is:

"The vertical length scale $(L_z)$ is specified a smaller constant value near the bottom and a larger constant value for the upper domain to be closer to the measured length scales, as explained in Xie and Castro (2008) and Veloudis et al. (2007)."

**Comment:** *15. Page 6, ": : :,explained in Xie and Castro (2008)". Please fix your grammar.*

**Response:** "explained in Xie and Castro (2008)" is replaced with "as explained in Xie and Castro (2008) and Veloudis et al. (2007)".

**Comment:** *16. Page 6, Line 14-15, "the vertically same wind direction", please fix your grammar.*

**Response:** "the vertically same wind direction" is replaced with "the constant wind direction vertically".

**Comment:** *17. Page 6, Line 19, "in the lateral direction", did you mean "spanwise" direction? Same for the rest of this paragraph. Lateral suggests both x- and y-directions.*

**Response:** "in the lateral direction" is replaced with "in the spanwise direction". This is also corrected in elsewhere of the manuscript.

**Comment:** *18. Fig. 1, caption, use "relative computation time" as in your main text, rather than "relative computation".*

**Response:** This is removed.

**Comment:** *19. Fig. 1, "dashed grey line of 1.0 indicating", indicates, not indicating.*

**Response:** This is removed.

**Comment:** *20. Page 7, Line 9, "filtered velocity" rather than "filter velocity".*

**Response:** "filter velocity" is replaced with "filtered velocity".

**Comment:** *21. Page 7, Line 14, and elsewhere. Please double-check on the GMD conventions, but I think you should spell out "Figure" if it is at the beginning of a sentence.*

**Response:** "Fig." is replaced with "Figure" all over the manuscript now, if it is at the beginning of a sentence.

**Comment:** *22. Fig. 2. Caption, "(b) The : : :" change to "(b) the: : :"*

**Response:** "The" is replaced with "the".

**Comment:** *23. Page 7, Line 16, "are advected and decay downwind: : :", please fix your grammar.*

**Response:** "are advected and decay downwind: : :" is replaced with "are advected in the domain: : :".

**Comment:** *24. Page 7, "can generate realistic well-configured turbulence structures from a short adjustment distance downwind". The adjustment distance does not look short to me. Judging from your Fig. 2b, it looks like a fetch distance of x = 5H is required at least. Please comment on this.*

**Response:** We have rephrased it to

"This suggests that the synthetic inflow turbulence generator can generate realistic well-configured turbulence structures from an adjustment distance downwind of about $x/H = 5\text{-}10$"

**Comment:** *25. Page 7, Line 21 to 22, "and there is no adjustment distance, and instead, an adjustment time to generate fully-developed turbulence structures". Please fix your grammar.*

**Response:** This sentence is removed now.

**Comment:** *26. Page 7, Line 28, "plan" or "plane"?*

**Response:** "plan" is replaced with "plane".

**Comment:** *27. Fig. 3, I suggest using the "global friction velocity $u_*$" from the periodic case to normalize $u_*$ for the inflow case. This way, we could detect the presence of systematic biases in the inflow case, if any.*

**Response:** Now the friction velocity for the inflow case in Fig 3 is scaled by the "global friction velocity" from the periodic case. The relevant modified text is as below:

"The variation of the local friction velocity is within $\pm 0.5\%$ $u_*$ along the streamwise direction for the periodic case and is slightly higher (within 1.5% $u_*$) than that for the inflow case after a downwind distance of $x/H = 7$."

**Comment:** *28. Fig. 3. caption "(laterally and temporally)", laterally and temporally averaged?*

**Response:** "laterally and temporally" is replaced with "the y-t averaged".

**Comment:** *29. Page 8, Line 10, "a good agreement against?" Please improve this sentence.*

**Response:** This is modified as follows:

"The horizontal profiles of normalised cross-stream velocity variance ($\langle v'^2 \rangle/u_*^2$ ) for the inflow case are in a good agreement after a developing distance of $x/H = 10\text{-}12$ , compared with these for the periodic case."

**Comment:** *30. Page 8, Line 12, can you comment on the possible reason for the slow convergence (long fetch distance) of wˆ2 at 0.1 z/H ?*

**Response:** This comment was for the figure in the first version. In the current version, as the profiles are smoother, we noticed that the difference is not evident. Therefore, we have revised this in the paper (Paragraph 1 of Section 3.1.3)

 "The development of normalised vertical velocity variance ($\langle w'^2 \rangle/u_*^2$) is achieved after a developing distance of about $x/H = 5\text{-}10$."

**Comment:** *31. Page 8, Line 15, why would "a larger shear-generated TKE" slow down the adjustment at 0.1z/H? Shouldn't this accelerate the adjustment because more TKE is generated locally independent of the TKE contained in the inflow.*

**Response:** See the above responses, e.g. the reply to Comment 30. This sentence is removed. The modified relevant text is:

"Since the streamwise velocity variance has a major contribution to TKE, the developing distance for TKE is similar to that for the streamwise velocity variance, i.e. about $x/H = 7\text{-}8$."

**Comment:** *32. Page 8, "downward turbulence transport from above" Did you look at the TKE budget? The transport term of TKE is quite insignificant in the NBL. Unless the inflow case is doing something less. It would be nice if you could present the TKE budgets and compare between the two cases.*

**Response:** See the responses for Comment G2 regarding this comment.

**Comment:** *33. Page 8, "The red circle dots", just "red circles" will do.*

**Response:** "The red circle dots" is replaced with "red line", to be consistent with new plots.

**Comment:** *34. Page 8, Line 21, "noticed again" or "noted again"?*

**Response:** "noticed again" is replaced with "noted again".

**Comment:** *35. Fig. 6, caption "$$ and $$ the laterally averaged mean and streamwise normal Reynolds stress", how are these Reynolds stresses? These are first-order moments.*

**Response:** $$ is replaced with $$ .

**Comment:** *36. Page 9, Line 17, "is able to sustained", please fix your grammar.*

**Response:** "is able to sustained" is replaced with "is able to be mostly sustained".

**Comment:** *37. Fig. 6, could you include a spectrum at the inlet x = 0, so that the readers have an idea of what the synthetic turbulence spectrum looks like?*

**Response:** The spectrum at the inlet $x = 0$ is added and the inertial subrange of -5/3 slope is shown in Fig. 6. The relevant modified text is:

"The spectrum drops slightly at high wavenumbers from the imposed spectra at $x/H = 0$ to downwind locations, and to recover towards the spectrum of the periodic case. The slight drop suggests a decay of small eddies due to the SGS and molecular viscosities"

**Comment:** *38. "A length scale (LS) ratio : : : are tested." Please fix your grammar.*

**Response:** "A length scale (LS) ratio : : : are tested." is replaced with "Length scale (LS) ratios : : : are tested."

**Comment:** *39. Page 9, bottom line "Fig. 8 (a) shows that $/u_*$ is slightly greater for the LS ratio less 1.0 (see Fig. 8a for comparison). This is due to a greater Reynolds shear stress $/u_*^2$. I do not understand your explanation. The velocity profile at z/H = 0.5 is affected by the divergence of the stresses, rather than the stress itself. How could a large stress value at z/H = 0.5 explain the overestimation of the velocity?*

**Response:** We are sorry that this was confusing. This has been revised to

"Figure 8 (a) shows that $\langle u \rangle / u_*$ is slightly greater for the length scale ratio less than 1.0. This is likely due to a slightly smaller $u_*$, which is common for smaller integral length scale cases (as shown in Fig. 7)."

**Comment:** *40. Page 10, Line 1, "Figs. 8(b-d) and (f)" rather than "Fig. 8(b-d)".*

**Response:** "Fig. 8(b-d) and (f)" is replaced with "Figures 8(b-d) and (f)".

**Comment:** *41. Page 10, Line 3, what is the "LE ratio"? did you mean your "LS 1.0" case?*

**Response:** Yes, it is fixed. "LE ratio equal to one" is replaced with "the LS 1.0 case".

**Comment:** *42. Page 10, Line 3, why is "LE ratio equal to one" the "accurate ones"? First of all, please fix your grammar. Second, what do you mean by "accurate"?*

**Response:** "the 'accurate' ones" is replaced with "the 'accurate' (compared with the periodic case) one". The 'accurate' is for the comparison to the periodic case.

**Comment:** *43. Page 10, Line 5, if all you have to say about Fig. 9 is that it "confirms the findings suggested from Fig. 8", I would suggest you remove that figure.*

**Response:** More discussion about Figure 9 is added on Paragraph 3 of Section 3.2:

"For $x/H = 10$, both mean and turbulent quantities converge approximately to the periodic case. In general, there are slight differences in $\langle u \rangle / u_*$ between each case. The magnitudes of turbulent quantities for smaller integral length scales are slightly smaller than those for larger integral length scales. This suggests that the mean velocity and the turbulent Reynolds stresses are not very sensitive to the integral length scales if they are not too different from the realistic values."

**Comment:** *44. Page 10, Line 9-10, "There is no significant change of the spectra", depends on what you mean by significant. The differences among these LS cases are similar to those presented in Fig. 6. I would suggest you plot your data on kE-log(k) plots. First, this avoids the flat 1D spectra issue at the low wavenumbers. Second, it would be much easier to tell the differences if the y-axis is not on a log scale.*

**Response:** Please see our reply to Comment M1. At x/H=10, all cases varying integral length scales generally converge to the periodic case with slight changes of the spectrum for small wavenumber turbulence. In the text, "no significant change" has been modified as "slight changes" for the new spectrum. There is no issue of flat spectra at the low wavenumbers for the new plots. In this paper, we present the spectrum plots with the inertial subrange of -5/3 slope (indicated in new plots), consistent with those in Xie and Castro (2008). The relevant text is modified (Paragraph 4 of Section 3.2)

"For all cases in the current study, the spectra with various integral length scales generally match those of the periodic case at a developing distance of $x/H = 10$, albeit with slight changes of the spectrum for small wavenumber turbulence. A very small variation of the spectra is within the uncertainty of the calculation of spectrum from the raw data. The spectra show an inertial subrange of -5/3 slope, which are consistent as those in the references, such as Xie and Castro (2008). "

"The spectrum in Munoz-Esparza et al. (2015) drops steeper at high wavenumbers, mainly due to a coarser resolution (noticing that their plots were for $kE_{u_i}$ with the inertial subrange of -2/3 slope). Our spectrum for $E_u$ has a broad range of the inertial subrange of -5/3 slope, indicated in Fig. 6."

**Comment:** *45. Page 10, Line 12, "idealised WRF-LES (v3.6.1) models", model not models*

**Response:** "idealised WRF-LES (v3.6.1) models" is replaced with "an idealised WRF-LES (v3.6.1) model".

**Comment:** *46. Page 11, Line 11, "The spectrum of these data shows an inertial subrange". I strongly recommend you show these in your spectra plots.*

**Response:** The inertial subrange  is now shown in the new spectrum  plots Figs. 6 and 10.

**Comment:** *47. Page 11, Line 12, "yields a satisfactory accuracy". Please, fix your grammar.*

**Response:** We have improved this statement, i.e.

"These tests on WRF also confirm that this method yields a satisfactory accuracy, after having compared *the local friction velocity, the mean velocity, the Reynolds stresses and the turbulence spectra* against the reference data."

---

## Author Comment (AC3) · 15 Feb 2020

**Responses to the comments from Anonymous Referee #3**

**General Remarks**

**Comment G1:** *The manuscript attempts to address a timely and relevant problem of inflow turbulence generation in large-eddy simulations of realistic atmospheric boundary layer flows. While there is nothing fundamentally wrong with the methodology applied the manuscript has a number of significant deficiencies. The review of previous work in the field is inadequate. The authors make several references to derived work instead of citing the original work (more details are provided under "Specific Remarks".)*

**Response:** We thank the reviewer for the critical comments, of which many are constructive.

This study attempts to equip WRF-LES with a well-tested synthetic turbulence inflow method (Xie and Castro 2008), which has been implemented and tested on engineering type of codes, such as Star-CD (Xie and Castro, 2009) and OpenFOAM (Kim and Xie, 2016) and the micro-scale meteorology code PALM (PALM, 2017; Maronga et al., 2019). This study can potentially provide a tool to bridge in WRF from mesoscale simulation (down to 1 km resolution) to the micro-scale Large-Eddy Simulations (tens of meters or less resolution) with additional turbulence information at small scales. In particular, we have highlighted the novelties in the revised paper, and have improved the review and citation of previous work in the introduction. More are detailed in the responses to "Specific Remarks".

**Comment G2:** *Only neutral boundary layer simulations are carried out and the Coriolis force was not activated. Such setup does not produce a realistic atmospheric boundary layer.*

**Response:** This study is focused on the feasibility of the inflow generation subroutine on WRF-LES using a periodic run as a control case. It is particularly focused on the sensitivity of the integral length scales on the turbulence development in the full-scale modelling of WRF under neutral atmospheric conditions. Turning off the Coriolis force is to achieve a constant wind direction vertically, enabling an easier interpretation of the impact of the integral length scales on the simulated flows. This kind of configuration (ignoring Coriolis force) has been used in previous work. A WRF-LES study by Ma and Liu (2017) removed the Coriolis force and used pressure gradient as the driving force to achieve a constant wind direction vertically for a simulation over a hill. Testing the Xie and Castro (2008) method for other conditions, such as considering the Coriolis effect, is out of the scope of this paper and can be the future work. Users of our open source subroutine may extend the code for their own applications.

**Comment G3:** *Furthermore, the synthetic turbulence generation approach of Xie and Castro (2008) was already implemented in WRF by Muñoz-Esparza et al. (2015), so it is not clear what is the original contribution of this work.*

**Response:** Munoz-Esparza et al. (PoF 2015) focused on their own developed and preferred method - the cell perturbation method, but not on the Xie and Castro (2008) method. To our best knowledge, their code of the Xie and Castro (2008) method has not been contributed as an open source. We made our inflow code (Xie and Castro, 2008) publicly available in this open source journal, i.e. Goescientific Model Development, which is one of the contributions to the community.

In addition, their numeric tests (Muñoz-Esparza et al., 2015) are based on the grid resolution of 90 m. The size of smallest eddy that can be resolved by the LES model is 180 m (i.e. twice the grid resolution). Given a boundary layer height of 500 m in their settings, there are just a small number of eddies resolved (considering the turbulence is anisotropic) in the vertical direction by their model. Our tests here adopt the grid resolution of 20 m. Munoz-Esparza et al. (2015) concluded that *the cell perturbation method*

*needs a fetch of 15-40 boundary layer depths to fully develop the turbulence, while the Xie and Castro (2008) method needs more fetch*. However, our conclusion in the current paper is that the Xie and Castro (2008) method only needs 5-15 boundary layer depths to fully develop the turbulence, and this is consistent with those in Xie & Castro (2008), Kim et al (2013) for engineering scale problems. This is obviously a new finding derived from a better configured model for the simulations of the full-scale atmospheric boundary layer than that in Muñoz-Esparza et al. (2015), although both use the Xie and Castro (2008) method implemented in WRF-LES.

Xie & Castro (2008) has been implemented in engineering type codes and is successful for wind-tunnel scale (ie. O(1m)) problems, but have not yet been tested rigorously in a meso-scale meteorological model. The focus of our current paper is to rigorously test and explore the Xie & Castro (2008) method in a full scale (i.g. very large Re number) problem, in terms of the sensitivity of integral length scales and the adjustment distance of the mean velocity field, the turbulent Reynolds stresses and the local friction velocity. Our paper will be extremely useful to the users of the Xie & Castro (2008) method in meso-scale meteorological models, such as WRF, and the micro-scale meteorology models, such as PALM. These are the novelties of the paper.

This work bridges the gap, such as in terms of the resolution (we use higher resolution than Munoz-Esparza et al (2014, 2015)), the sensitivity of the turbulent statistics due to the change of integral length scales, for a systematic study of the properties of the method in the WRF-LES model. Of course, we are not able to address everything in this aspect. We also do agree with the authors that Munoz-Esparza et al (2014, 2015)) provide an alternative for turbulence generation for such applications.

**Comment G4:** *Finally, some of the conclusions about the effectiveness of the synthetic turbulence generation approach are not supported by the results presented in the manuscript. In particular, the length of the fetch needed to achieve the equilibrium boundary layer is underestimated.*

**Response:** In response to the reviewer's comment, we have conducted more detailed analyses, including re-postprocessinmg more model output (i.e. using the higher-frequency output - every 5 sec in contrast to every 1 min in the previous analysis) to generate better turbulence statistics and spectra (see the Figs. 5, 6, 9, and 10, for example). These have largely helped us to make more solid conclusions on the effectiveness of the synthetic turbulence generation approach. See more specific replies to the specific remarks.

The length of the fetch needed to achieve the equilibrium boundary layer has been carefully assessed for the turbulent Reynolds stresses, TKE and the local friction velocity. Our conclusion in the current paper is that the Xie and Castro (2008) method needs 5-15 boundary layer depths to fully develop the turbulence, and this is consistent with those in Xie & Castro (2008), Kim et al (2013) for engineering scale problems. For a coarser grid resolution of 90 m (vs 20 m in our paper), Munoz-Esparza et al. (2015) concluded that "the cell perturbation method needs a fetch of 15-40 boundary layer depths to fully develop the turbulence, while the Xie and Castro (2008) method needs more fetch". We speculate it is mainly because Munoz-Esparza et al. (2015) used a much coarser mesh in their tests.

We have added/modified the following related text:

"Since the streamwise velocity variance has a major contribution to TKE, the developing distance for TKE is similar to that for the streamwise velocity variance, i.e. about $x/H = 7$-8. The distance needed for different quantities to reach a converged state differs from each other, and it is about $x/H = 5$-15."

**Comment G5:** *Taking all the above into account I do not recommend the manuscript for publication in the journal Goescientific Model Development.*

**Response:**  Anyway, we have taken the reviewer's critical (including many constructive) points. We would like to cite here some points from the other two reviewers:

*"this work deserves to be published since it involves a rather systematic study of the properties of the method in the WRF-LES model. Especially, the sensitivity study to the integral length scale provides some new and very likely useful information."*

*"The gray zone between the scales resolvable by the meso-scale models and the resolution requirements of LES unavoidably lead to a large gap in the resolution and therefore it becomes very important to somehow incorporate the lacking turbulence information on the inflow boundaries of the LES-domain in some more or less approximative manner."*

*"This work will benefit the atmospheric community by providing then with a practical engineering tool for improving nested simulations at the LES scale. Implementing a piece of code like this into WRF is no "a walk in the park", it must have taken the authors a great deal of time and effort. For that I appreciate their efforts, and applaud them for making their code publicly available with this manuscript."*

We have carefully addressed the major concerns raised by the reviewer, and also improved the manuscript by addressing the specific remarks suggested by the reviewer as below.

**Specific Remarks**

**Comment:** *Page 2, line 2 – The reference to Nottrott et al. is not appropriate, since Nottrott et al. did not develop WRF. Proper reference would be Skamarock and Klemp (JCP 2008).*

**Response:** We thank the reviewer for the suggestion. This reference Skamarock and Klemp (JCP 2008) was cited when the WRF model was mentioned in our previous version (i.e. in the sentence before Page 2, line 2). Here, the reference to Nottrott et al. is now replaced with "(Skamarock and Klemp, 2008)".

**Comment:** *Page 2, line 7 – Doubrawa et al. 2018 is certainly not the first or most important*

*reference related to WRF-LES.*

**Response:** Doubrawa et al. 2018 is a study on the downscaling from mesoscale simulation to the LES, i.e. linked to the terra incognita range of grid resolutions. More related references are added in the revised paper, i.e. "Doubrawa et al., 2018; Talbot et al., 2012; Chu et al., 2014; Liu et al., 2011".

**Comment:** *Page 2, line 11 – This is not an example of a fundamental study. Nunalee et al. (2014) reported on LES using WRF model based on a tracer dispersion field study and compared simulation results to field study observations.*

**Response:**  We thank the reviewer for the comment. In response to the comment, we've added the first study of testing nested LES in WRF by Moeng et al. (2007) and other relevant studies in the revised paper. The word of "fundamental" is removed and more details about studies (with some references) are added as follows,

"Therefore such periodic WRF-LES simulations are restricted to *studies* of the atmospheric boundary layer flow *with a single domain (e.g. Zhu et al., 2016; Kirkil et al., 2012; Kang and Lenschow, 2014; Ma and Liu, 2017) or the outermost domain for the nested cases (e.g. Moeng et al., 2007; Khani and Porte-Agel, 2017; Nunalee et al., 2014).*"

Nunalee et al. (2014) used periodic conditions for the parent domain in the nested WRF-LES simulations for the tracer dispersion study and also compared meteorological conditions (i.e. hourly mean vertical profiles of wind speed, potential temperature and wind direction in their Fig. 4) with the field measurements. Nunalee et al. (2014) is kept in the revised paper as an example case for the use of periodic conditions for the parent domain in nested WRF-LES cases.

**Comment:** *Page 2, line 14 – Munoz-Esparza et al. (PoF 2015) have already implemented synthetic turbulence inflow scheme by Xie and Castro (2008), so it is not clear what is the original contribution of this work.*

**Response:** This is also commented in Comment G3. See our responses to Comment G3.

**Comment:** *Page 2, line 20 – A space is missing between year and semicolon, here, and on numerous places throughout the manuscript.*

**Response:** This is due to the formatting of Endnote for multiple citations.  A space is added between multiple citations and this has been checked throughout the manuscript.

**Comment:** *Page 2, line 26 – However, the velocity profile could be modified, also it can vary in time.*

**Response:** These is for the discussion of the library-based method and recycling-rescaling based method, which are normally applicable to the idealised LES simulations of stationary and equilibrium flows. We have added the following here,

"The turbulence profile determined by the geometry of the precursor simulation can be added on the top of any given mean profile, which could be modified and varied in time for more realistic applications."

**Comment:** *Page 3, line 16 – More recent reference that expands and improves on Muñoz-Esparza et al. (2015) is Muñoz-Esparza and Kosovic (2018).*

**Response:** The more recent reference is added as follows:

"Munoz-Esparza and Kosovic (2018) extended the cell perturbation method of the inflow turbulence generation to non-neutral atmospheric boundary layers."

**Comment:** *Page 4, line 4 – A subgrid scale scheme does not parameterize small unresolved eddies, instead it parameterizes the effect of small unresolved eddies on the resolved field.*

**Response:** This is modified as follows: "which computes large energy-containing eddies at the resolved scale directly and parameterises the effect of small unresolved eddies on the resolved field using subgrid-scale (SGS) turbulence schemes (Moeng et al., 2007)."

**Comment:** *Page 5, Equation 9, 12, etc. – The notation using plus sign is confusing since subscript*

*m indicates the velocity component.*

**Response:** We are sorry that the reviewer was confused here because we used '*m*' to index two different quantities by mistake. We have now corrected it. In the revised manuscript, we've added "*m*, the index that the averaging operator is applied, denotes the *m*-th element of a vector (one-dimensional data series of, for example, the digital-filtered velocity, *u*, in (9) below), $k$ is the number of elements for the two-point distance of $k\Delta x$" for explanation when these first appear in Eq. (8). In addition, "*m*" in Eqs. (12) and (13) is replaced by "β" to indicate velocity components. "*j*" in Eqs. (14) and (15) is also replaced by "β". These notations follow those adopted by Xie & Castro (2008).

**Comment:** *Page 6, line 15 – Why is Coriolis turned off if simulation of flow in an atmospheric boundary layer is the goal?*

**Response:** This is also commented in Comment G2. See the responses to Comment G2.

**Comment:** *Page 7, line 6 – Doubling the computational time is a significant increase that needs to be justified.*

**Response:** This is due to that "the additional computational time associated with subroutine of the synthetic inflow turbulence generator and data passing, which is not parallelised, while the main code WRF is parallelised".

We emphasize again that this study is focused on the feasibility of implementing the inflow method (Xie & Castro, 2008) in the meso-to-micro-scale meteorological code of WRF and the impact of the key variables (i.e. the integral length scales) on the simulated turbulence development inside the domain. Up to the authors' knowledge, the latter has not been rigorously addressed previously. We appreciate that *the technical parallelisation of the Xie & Castro (2008) method has been done in PALM and some other researchers (e.g. Kim and Xie, 2016) have also made efforts to technically parallelise the Xie & Castro method. These suggest that technically parallelising this method is not an issue. It is our intention that we test the method inside WRF scientifically and rigorously here and publish our open source subroutine through GMD to allow other WRF-LES users to extend technical capabilities of our code, such as parallelisation.* A paragraph in the Discussion and conclusions section is added for the discussion about the parallelisation of the method, i.e.

"This study is focused on the feasibility of implementing the inflow method (Xie & Castro, 2008) in the meso-to-micro-scale meteorological code WRF and the impact of the key variables (i.e. the integral length scales) on the simulated turbulence development inside the domain. This inflow subroutine has previously been implemented in both serial and parallel mode in several codes, including engineering type of codes Star-CD (Xie and Castro, 2009) and OpenFOAM (Kim and Xie, 2016), and the micro-scale meteorology code PALM (PALM, 2017). Although the current implementation in WRF is affordable for a moderate-sized simulation (e.g. tens of meters resolutions), the technical parallelisation of this inflow subroutine in WRF-LES can be the future work for very large simulation domains with high resolutions. Users of our open source subroutine may offer this technical contribution."

**Comment:** *Page 7, line 7 – The adjustment distance should be more precisely quantified.*

**Response:** "about $x/H = 5\text{-}10$" is added in the revised paper.

**Comment:** *Page 8, line 2 – Instead of symbols, the stresses should be defined as: "horizontal profiles of normal and shear turbulent stresses normalized by surface friction velocity."*

**Response:** Symbols have been removed. The following text is added, i.e.

"horizontal profiles of normalised mean streamwise velocity component, normal and shear turbulent stresses, and TKE".

**Comment:** *Page 8, line 7 – Normalized streamwise variance matches well at x/H = 7 or 8 and not at x/H = 5.*

**Response:** "$x/H = 5$" is replaced with "$x/H = 7\text{-}8$".

**Comment:** *Page 8, line 10 – Below z/H = 0.3 the profile of cross-stream variance differs significantly for any x/H.*

**Response:** It has been modified as follows:

"The horizontal profiles of normalised cross-stream velocity variance ($\langle v'^2 \rangle / u_*^2$ ) for the inflow case are in a good agreement after a developing distance of $x/H = 10\text{-}12$ , compared with those for the periodic case."

**Comment:** *Page 8, line 12 – The development is not achieved at all, since only at the end of the domain the values of <w'2>/u*2 obtained using the synthetic turbulence generation method are the same as those from the simulation involving periodic domain. Also, what is shown in the figures is the fetch, not the time scale.*

**Response:** This comment was for the figure in the first version. In the current version, as the profiles are smoother, we noticed that the difference is not evident. Therefore, we have revised this in the paper.

"The development of normalised vertical velocity variance ($\langle w'^2 \rangle / u_*^2$) is achieved after a developing distance of about $x/H = 5\text{-}10$."

The time scale has been changed to "length scale".

**Comment:** *Page 8, line 13 – Figures show that the fetch needed for different quantities to reach the equilibrium values differs significantly between them. For example, vertical velocity variance does not reach equilibrium. Since it is a component of TKE, TKE also requires a long fetch to reach the equilibrium.*

**Response:** See our reply to the Comment G4. We have regenerated and re-examined these plots carefully. Based on these smoother profiles, we are able to reach more solid conclusions. These plots suggest that the *fetch needed for different quantities to reach the equilibrium values differs only slightly between them, considering a small uncertainties (errors) due to the limited averaging time.*

The fetch needed for different quantities is also discussed in the responses to Comment G4.

**Comment:** *Page 8, line 18 – Same as above, these should be labeled as normal and shear turbulent stresses normalized by surface friction velocity.*

**Response:** Symbols have been removed. The following text is added, i.e.

"normalised mean streamwise velocity component, normal and shear turbulent stresses, and TKE".

**Comment:** *Page 8, line 22 – A sentence should not begin with a symbol.*

**Response:** "The normalised mean streamwise velocity component" is added before the symbol.

**Comment:** *Page 8, line 23 – In "matches closely to that: : :," "to" should be omitted.*

**Response:** "to" is removed.

**Comment:** *Page 8, line 24 – Same as above, instead of symbols names of the terms should be used.*

**Response:** This is modified as "The normalised streamwise velocity variance".

**Comment:** *Page 9, line 9 - The spectral roll-off depends on the numerics not on the turbulence generation scheme, so this is questionable conclusion. Also, flat spectra over a decade of wave numbers is not realistic. Furthermore, there is not apparent inertial range (-5/3) slope in the results presented in Figure 6.*

**Response:** In the previous version, we conducted spectral analysis using the spatial data along the cross-stream direction ($y$) with given values of $x$ ($x/H$=2, 4, 6 and 10) and $z$ (=0.5$H$) and then averaged the spectrum over time to generate the results in Figure 6 in the original manuscript. Now a different method is adopted: for given values of $x$ and $z$, we conduct spectral analysis using the time series of 3600 s with an interval of 5 s for five selected sample locations of $y_n$ ($y/H =$ 1.76, 2.16, 2.56, 2.96 and 3.36), namely, $\tilde{u}(t, 2H, y_n, 0.5H)$, and then an average over $y_n$ yields the data plotted in Fig. 6 in the revised manuscript. This method is in essence used to analyse experimental time series data from point measurement; when applied to the LES data, it yields a fairly good inertial subrange as shown in the new Figure 6, as well as in new Figure 10.

We have modified and added the following (in Section 3.1.5):

"For each $x$-location, e.g. $x/H = 10$, the spectrum for the inflow case was firstly calculated from the streamwise wind velocity component over a time series of 3600 s with an interval of 5 s for five selected sample locations of $y_n$ ($y/H = 1.76, 2.16, 2.56, 2.96$ and $3.36$), namely, $\tilde{u}(t, 2H, y_n, 0.5H)$. The spectral data were then averaged over $y_n$ to give the spectra plotted in Fig. 6. "

"The spectrum drops slightly at high wavenumbers from the imposed spectra at $x/H = 0$ to downwind locations, and to recover towards the spectrum of the periodic case. The slight drop suggests a decay of small eddies due to the SGS and molecular viscosities"

"The spectrum in Munoz-Esparza et al. (2015) drops steeper at high wavenumbers, mainly due to a coarser resolution (noticing that their plots were for $kE_{u_i}$ with the inertial subrange of -2/3 slope). Our spectrum for $E_u$ has a broad range of the inertial subrange of -5/3 slope, indicated in Fig. 6."

**Comment:** *Page 9, line 14 – If current work does not differ from Munoz-Esparaza et al. (2015), what is new in the present manuscript?*

**Response:** This is also raised in Comment G3. See the responses to Comment G3.

**Comment:** *Page 9, line 24 – Instead of "slightly affects," it should be "affects slightly."*

**Response:** As suggested, this is now corrected.

**Comment:** *Page 9, line 30 – As before, words should be used instead of symbols.*

**Response:** Symbols have been removed. The following text is added, i.e.

"normalised mean streamwise velocity component, normal and shear turbulent stresses, and TKE".

**Comment:** *Page 10, line 3 – It is not clear what is meant by " 'accurate' ones..."*

**Response:** "the 'accurate' ones" is replaced with "the 'accurate' (compared with the periodic case) one". The 'accurate' is for the comparison to the periodic case.

**Comment:** *Page 10, line 16 – It is not clear what is the purpose of the statement starting with "It is not trivial: : :" This statement by itself is of little relevance, the question is: What is the relevance?*

**Response:** This has been modified:

"It is not trivial to estimate the integral length scales as the primary input of the inflow turbulence generator." is replaced with "It is important to estimate the integral length scales, which are the key inputs of the inflow turbulence generator."

**Comment:** *Page 10, line 21 – The adjustment fetch should be quantified. It is not short.*

**Response:** This is now quantified, i.e. "after a short adjustment distance" is replaced with "after an adjustment distance of $x/H$=5-15".

**Comment:** *Page 11, line 12 – The statement related to ": : :a satisfactory accuracy" is an imprecise qualitative statement. It should be stated what is the accuracy satisfactory in comparison to.*

**Response:** We have improved this statement, i.e.

"These tests on WRF also confirm that this method yields a satisfactory accuracy, after having compared *the local friction velocity, the mean velocity, the Reynolds stresses and the turbulence spectra* against the reference data."

---

## Referee Report (RR1)

**Review report**

Having reviewed the original manuscript, and now after reviewing the revised manuscript, I can recommend its acceptance for publication subject to minor revisions.

Generally, the english language would still need improvements here and there. The following comments must be taken into account and appropriate corrections should be made. In the following PXLY means page X line Y.

P1L30: "mesoscale scales" → mesoscales

P2L28: "…from a downstream boundary back to the upstream inlet." It makes no sense to recycle the velocity field from the downstream boundary. This would be like in the normal cyclic boundary condition. Normally the data to be recycled is taken from some downstream plane between the inflow boundary and the principal area of interest. So, this could be written e.g. as: "...from some suitably selected downstream plane back to the inflow boundary plane."

P7L21: "wass"  (please note that there are typos elsewhere, too)

P9L4: "shear turbulent stress" → turbulent shear stress

P10L6: "...due to the SGS and molecular viscosities." Frankly, the molecular viscosity plays no role at this high Reynolds number and this grid resolution. The SGS viscosity certainly exceeds it everywhere by several orders of magnitude. However, the numerical dissipation originating from the fifth-order upwind-biased Wicker-Skamarock advection scheme used in WRF is probably of comparable importance in the spectrum high-end drop as is the SGS-dissipation. This should be mentioned.

P11L30-32: some questionable language: "...consistent with in the findings..." and "...may be owe to the...".

P12L27: "…shows a broad inertial subrange of -5/3 slope." In my view the inertial subrange in the spectra shown in Figs. 6 and 10 is not at all broad. Instead, it is just hardly distinguishable. The word broad must be erased here.

P25L13-18: Reference Maronga et al 2019 is outdated. That article has been published in Geosci. Model Dev., 13, 1335–1372, 2020 https://doi.org/10.5194/gmd-13-1335-2020. The reference must be updated accordingly.

---

## Author Response (AR2)

**Responses to the comments from Anonymous Referee #1**

**General comments**

*Having reviewed the original manuscript, and now after reviewing the revised manuscript, I can recommend its acceptance for publication subject to minor revisions.*

*Generally, the english language would still need improvements here and there. The following comments must be taken into account and appropriate corrections should be made. In the following PXLY means page X line Y.*

**Response:** We thank the Reviewer for reviewing our manuscripts and having given very positive comments. We have read the manuscript thoroughly and have improved the English with the help of a native speaker. We have highlighted our changes in the revised manuscript. We respond to each of the Reviewer's specific comments below.

**Comment**: *P1L30: "mesoscale scales" → mesoscales*

**Response:** This has been corrected.

**Comment**: *P2L28: "...from a downstream boundary back to the upstream inlet." It makes no sense to recycle the velocity field from the downstream boundary. This would be like in the normal cyclic boundary condition. Normally the data to be recycled is taken from some downstream plane between the inflow boundary and the principal area of interest. So, this could be written e.g. as: "...from some suitably selected downstream plane back to the inflow boundary plane."*

**Response:** As suggested, this has been replaced with "...from some suitably selected downstream plane back to the inflow boundary plane."

**Comment**: *P7L21: "wass" (please note that there are typos elsewhere, too)*

**Response:** This has been corrected. We have read the manuscript thoroughly and have improved the English with the help of a native speaker. We have highlighted our changes in the revised manuscript.

**Comment**: *P9L4: "shear turbulent stress" → turbulent shear stress*

**Response:** This has been corrected.

**Comment**: *P10L6: "...due to the SGS and molecular viscosities." Frankly, the molecular viscosity plays no role at this high Reynolds number and this grid resolution. The SGS viscosity certainly exceeds it everywhere by several orders of magnitude. However, the numerical dissipation originating from the fifth-order upwind-biased Wicker-Skamarock advection scheme used in WRF is probably of comparable importance in the spectrum high-end drop as is the SGS-dissipation. This should be mentioned.*

**Response:** This has been replaced with "due to the SGS viscosities and the numerical dissipation originating from the advection scheme in the WRF-LES model".

**Comment**: *P11L30-32: some questionable language: "...consistent with in the findings..." and "...may be owe to the...".*

**Response:** These have been replaced with "agrees with the findings in" and "can be attributed to".

**Comment**: *P12L27: "...shows a broad inertial subrange of -5/3 slope." In my view the inertial subrange in the spectra shown in Figs. 6 and 10 is not at all broad. Instead, it is just hardly distinguishable. The word broad must be erased here.*

**Response:** As suggested, the word broad has been erased. This has been replaced with "shows an inertial subrange of -5/3 slope".

**Comment**: *P25L13-18: Reference Maronga et al 2019 is outdated. That article has been published in Geosci. Model Dev., 13, 1335–1372, 2020 https://doi.org/10.5194/gmd-13-1335-2020. The reference must be updated accordingly.*

**Response:** This has been updated accordingly.

**Responses to the comments from Anonymous Referee #3**

**Comment**: *Review of the revised manuscript GMD-2019-165: "Implementation of a synthetic inflow turbulence generator in idealised WRF v3.6.1 large eddy simulations under neutral atmospheric conditions" by Jian Zhong, Xiaoming Cai[1] and Zheng-Tong Xie submitted for publication in the journal Geoscientific Model Development.*

*In the revised manuscript "Implementation of a synthetic inflow turbulence generator in idealised WRF v3.6.1 large eddy simulations under neutral atmospheric conditions" the authors have made relatively minor changes to the exposition of the methodology for generation of inflow turbulence for large-eddy simulations based on synthetic turbulence generation approach developed by Xie and Castro (2008) implemented in the Weather Research and Forecasting model. In addition, the results presented in the revised manuscript are improved in comparison to the original manuscript.*

**Response:** We thank the Reviewer for the positive comments on the improved methodology and the results. Below we respond to all of the Reviewer's comments.

**General Remarks**

**Comment**: *While, the revised manuscript includes improved results, the main deficiency of the manuscript remains – the synthetic inflow turbulence generation methodology developed by Xie and Castro (2008) was previously implemented in the Weather Research and Forecasting model by Muñoz-Esparza et al. (2015) and extensively tested. It is therefore not clear what new research is presented in this manuscript. One element that is explored in greater detail is integral length scale, however, the authors do not clearly articulate any potential differences in implementation or results of simulations. Furthermore, the authors claim that the simulations presented in the manuscript are of neutral atmospheric conditions, however, Coriolis force was not used in the simulations and therefore important characteristic of a real atmospheric boundary layer, namely wind veering, could not be reproduced. Finally, although minor changes to the background material and exposition were introduced, some of these include misrepresentation of previous work. Examples are given below under Specific Remarks.*

*Taking all the above into account I do not recommend the manuscript for publication in the journal Goescientific Model Development.*

**Response:** As we emphasized in our previous responses to the Reviewer, here we repeat our points again. Muñoz-Esparza et al. (PoF 2015) focused on their own developed and PREFERED method - the cell perturbation method, but not on the Xie and Castro (2008) method, for the mesoscale to microscale transition. In addition, to the best of our knowledge, their subroutine code with the Xie and Castro (2008) method has not been contributed as an open source, whereas we are making our inflow code (Xie and Castro, 2008) publicly available in this open source journal, i.e. Goescientific Model Development, which is one of the contributions to the community.

The tests in Muñoz-Esparza et al. (2015) are based on the horizontal grid resolution of 90 m and the size of smallest eddies resolved by the LES model is about 180 m (i.e. twice the grid resolution). Considering their boundary layer height of $z_i \cong z_{i0} = 500$ m, this horizontal resolution is insufficient to represent the most energetic turbulent eddies (near $z/z_i \sim 0.16$ in their Fig. 7a) in the neutral boundary layer, specifically the vertical component of resolved turbulence. This is illustrated by their spectra plots for $z/z_i \sim 0.1$ of Fig. 9: the w-spectra are about one order of magnitude smaller than the v-spectra, even

for the periodic B.C. setting. Our tests here adopt the grid resolution of 20 m, providing a much finer mesh to resolve more eddies for the same boundary height of 500m; for example, our Fig. 5(d) shows that the magnitude of the w-component resolved energy is a large fraction of that of the v-component. Therefore, our results are more reliable on testing Xie and Castro (2008) method in WRF.

Muñoz-Esparza et al. (2015) concluded that *the cell perturbation method needs a fetch of 15 boundary layer depths to fully develop the turbulence, while the Xie and Castro (2008) method needs more fetch*. However, our conclusion in the current paper (with much higher grid resolution in WRF) is that the Xie and Castro (2008) method in WRF only needs 5-15 boundary layer depths to develop turbulence to the required level, and this is consistent with those findings in Xie & Castro (2008) and Kim et al (2013) for engineering scale problems. This is certainly a novel finding derived from a better configured model for the simulations of the full-scale atmospheric boundary layer than that in Muñoz-Esparza et al. (2015), although both implemented independently the Xie and Castro (2008) method in WRF-LES.

Xie & Castro (2008) has been implemented in engineering-type codes and is successful for wind-tunnel scale (ie. $O(1m)$) problems, but have not yet been tested rigorously in a meso-scale meteorological model. The focus of our current paper is to rigorously test and explore the Xie & Castro (2008) method in a full scale (i.e. in flow with a very large Re number) problem, in terms of the sensitivity of integral length scales and the adjustment distance of the mean velocity, the turbulent stresses and the local friction velocity. We found and emphasized in the revised manuscript that "*The mean velocity profiles at all tested locations were in very good agreement with the reference data, while the turbulence second moment statistics profiles were in reasonable agreement with the reference data about x/H=5-15 downwind of the inlet. An accurate estimation of the second order moments are crucial for the assessment of the synthetic inflow turbulence generator, in particular when the inflow turbulence information is not completely available. We found varying the integral length scale within +/-40% of the value in the base case has a negligible influence on the mean velocity profiles, while the effects of the variation on the turbulent second order moment statistics are visible, for example the local friction velocity was within 4 % error of the reference data at x/H=7.*" Our paper will be extremely useful to the users of the Xie & Castro (2008) method in meso-scale meteorological models, such as WRF, and the micro-scale meteorology models, such as PALM. This is another novelty of the paper in terms of the use of turbulence integral length scale.

For the Reviewer's comment on the Coriolis force, again as we emphasized in our previous revision:

Turning off the Coriolis force is to achieve a constant wind direction in the vertical direction, enabling an easier interpretation of the impact of the integral length scales on the simulated flows in this study. Such a configuration (where the Coriolis force is not activated) has been used in previous WRF-LES studies too (e.g. Ma and Liu 2017). *It is worth, however, a future study to examine the wind spiral case induced by the Coriolis force in the atmospheric boundary layer* (This has been added in the Section 4 Discussion and conclusions of the revised manuscript).

Further changes to the discussion of previous work are given in the responses to the Reviewer's specific remarks, below.

**Specific Remarks**

**Comment**: *Page 2, line 13 – It is stated that: "Therefore such periodic WRF-LES simulations are restricted to studies of the atmospheric boundary layer flow with a single domain (e.g. Zhu et al., 2016; Kirkil et al., 2012; Kang and Lenschow, 2014; Ma and Liu, 2017) or the outmost domain for the nested cases (e.g. Moeng et al., 2007; Khani and Porte-Agel, 2017; Nunalee et al., 2014)." However, the*

*simulation listed here are quite different. While Moeng et al. (2007) present simulations included two-way nested domains where periodicity on the outer domain impacts the flow on the inner domain, Nunalee et al. (2014) present a one-way nested simulations where inner domain is not impacted by periodicity on the outer domain, so that such setup can be used for simulation of heterogeneous boundary layers.*

**Response:** The boundary condition for the outer domain of the nested cases is periodic, while the inner domain can be either one-way nested (e.g. Nunalee et al. 2014) or two-way nested (e.g. Moeng et al. 2007). This has been further clarified in the revised manuscript, as follows:

"… or the outermost domain of either one-way nested cases (e.g. Nunalee et al. 2014) or two-way nested cases (e.g. Moeng et al., 2007)."

**Comment**: *Page 3, line 21 – It is stated that: "Generally speaking, these methods impose "white-noise" perturbations, thus having a flat spectrum, to a variable (e.g. temperature) at the inlet, and the model dynamics will "process" the signals once these signals are advected into the domain, e.g. to dissipate high-wavenumber signals quickly and to adjust low-wavenumber signals gradually." This statement misrepresents the temperature perturbation methodology. Temperature perturbations are introduced at a specific length and time scale related to the highest well resolved wave number in LES and therefore they cannot be considered "white noise." White noise can be defined as "random signal having equal intensity at different frequencies, giving it a constant power spectral density."*

**Response:** The "white-noise" has been removed. As suggested, this sentence has been changed as follows:

"These methods impose temperature perturbations at specific length and time scales related to the highest resolved wave number in the LES"

**Comment**: *Page 3, line 25 – It is stated: "It is thus not surprising that a large distance of about 20-40 boundary-layer depths (Muñoz-Esparza et al., 2015; Mazzaro et al., 2019) is normally required to allow a transition to fully-developed turbulence." However, Muñoz-Esparza et al. (2015) demonstrated that "From those results, it is evident that the performance of the cell perturbation method is not affected by these factors, rather induces turbulent structures that become fully developed at x/zi0 ≈ 15." See also Figure 18 in Muñoz-Esparza et al. 2015.*

*Furthermore, the difference between the application of synthetic inflow turbulence generator presented in the manuscript and the temperature perturbation methodology presented in Muñoz-Esparza et al. (2015) is not recognized by the authors. Temperature perturbation is introduced for mesoscale to microscale coupling approach where smooth mesoscale flow (no resolved turbulence) is forcing microscale flow and for that purpose one-way nesting approach is used in WRF. The nesting approach necessarily represents a different challenge for transition to fully-developed turbulence due to nesting compared to just specifying inflow turbulence, e.g., there is significant difference in inflow turbulence levels between the results presented in the manuscript and those in Muñoz-Esparza et al. (2015).*

**Response:** We have changed this in the revised manuscript accordingly:

"It was demonstrated in Muñoz-Esparza et al. (2015) that a distance of about 15 boundary-layer depths is required to allow the flow to be fully turbulent when the temperature perturbation method is adopted in the one-way nesting WRF model. Noted that the temperature perturbation method was introduced for mesoscale to microscale coupling approach where smooth mesoscale flow (no resolved turbulence) forces microscale flow by using the one-way nesting approach in WRF. Muñoz-Esparza et al. (2014)

stated "the perturbation method is to provide a mechanism that accelerates the transition towards turbulence, rather than to impose a developed turbulent field at the inflow planes as the synthetic turbulence generation methods pursue", and "the use of temperature perturbations presents an alternative to the classical velocity perturbations commonly used by most of the techniques"."

**Comment**: *Page 10, line 6 – It is stated: "The spectrum in Muñoz-Esparza et al. (2015) drops steeper at high wave numbers, mainly due to a coarser resolution…" The drop in the spectra is related to the implicit filter associated with the numerical discretization and drops at lower wave numbers due to coarser resolution, but it does not drop steeper. Steepness of the drop is related to the implicit filter. It has been determined that the effective resolution of WRF is ~7 dx (Skamarock 2004).*

**Response:** Thank the reviewer for clarifying this. We have changed this in the manuscript accordingly:

"The spectra in Muñoz-Esparza et al. (2015) drop at lower wave numbers than those in Fig. 6, mainly due to a coarser resolution (than the current one)."

---

## Author Response (AR3)

**Responses to the comments from Editor**

*Comments to the Author:*

*I have only a small number of very minor further comments:*

**Response:** We appreciate the editor's prompt reply and the minor comments.

*Comment:* * The code, referenced in the "Code and data availability" section, should appear in the bibliography and be cited here (see the code and data policy).*

**Response:** The reference for the code is added in the bibliography and is cited in "Code and data availability" section.

*Comment:* * I am not sure that the sentence "Users of our open source subroutine may offer this technical contribution." is needed.*

**Response:** It has been removed at the bottom of Page 12.

*Comment:* Typos etc:*

** Page 3 line 25: "Note" instead of "Noted"?*

**Response:** We are sorry for the typo. This has been replaced with "It is to be noted that".

*Comment:* * Page 10 line 14: Double full stop.*

**Response:** We are sorry for the typo.  It has been amended.